# Anchored Self-Play for Code Repair

**Caroline Choi** [1]  **Zeyneb Kaya** [1]  **Shirley Wu** [1]  **Tengyu Ma** [1]  **Tatsunori Hashimoto** [1]  **Ludwig Schmidt** [1]

## Abstract

Code repair is an important capability for language models (LMs): given a buggy program and unit tests, an LM must produce a fixed program that passes the tests. Because code repair data is limited, we aim to scale supervision by using an LM to generate bug–fix tasks via unconstrained edits and validating them only through unit test outcomes. We propose *generator–fixer self-play*, in which a single model is trained with reinforcement learning to generate bugs and fix them. As the fixer improves, the generator adapts to produce increasingly difficult bugs, yielding an automatic curriculum. To assess whether generator–fixer self-play generalizes beyond its own synthetic bugs, we introduce BUGSOURCEBENCH, which evaluates repair across realistic bug sources: human-authored bugs, errors in LM-generated code, and human edits of buggy LM-generated code. On BUGSOURCEBENCH, we find that self-play can drift toward difficult but unrealistic bugs, improving on self-generated bugs while degrading on human-written bugs. We propose ANCHORED SELF-PLAY (ASP), which anchors self-play with a small reference set by (i) adding a code-embedding similarity reward to guide generation and (ii) mixing reference bugs into fixer training to prevent drift. Across bug sources, ASP achieves the best fix rates, improving average fix rate by $+25\%$ relative / $+7.2$ pp absolute over standard self-play, with gains on both LM-error bugs and human-authored bugs.

## 1. Introduction

As language models (LMs) are increasingly used in programming workflows, reliable code repair has become an

important capability (Xu et al., 2022; Jimenez et al., 2023). Given a buggy program and accompanying unit tests, the LM must produce a correct program that passes the tests. However, obtaining large collections of realistic buggy programs is costly and hard to scale (Just et al., 2014; Widyasari et al., 2020; Le Goues et al., 2015; Madeiral et al., 2019; Oliva et al., 2025).

We ask whether we can scale supervision for code repair by *adaptively* generating buggy programs, using unit tests as verification. Specifically, we study an *open-ended* setting in which the bug generator can apply arbitrary text edits, rather than pre-defined mutations or repository-history rewrites, so synthetic training data is not limited to a fixed set of tasks. As the fixer improves, the generator should adapt to produce increasingly challenging yet realistic bugs, yielding an automatic curriculum.

We operationalize this idea with *generator–fixer self-play* (Figure 1). A single model is trained with reinforcement learning to alternate between (i) generating a bug via unconstrained text edits and (ii) fixing the bug. The generator is rewarded for producing valid, appropriately difficult bugs (tests fail), and the fixer is rewarded for producing correct repairs (tests pass). This creates an adaptive curriculum: as the fixer improves, the generator must generate harder bugs.

The main challenge in generator–fixer self-play is distribution drift. Unit tests certify correctness but not realism: many unconstrained edits can break tests, but few resemble bugs encountered in practice. As training progresses, the generator can drift toward unnatural yet test-failing edits, improving repair on self-generated bugs while degrading on real-world bugs. Concurrent work on self-play for software engineering agents (Wei et al., 2025) and prior work on bug generation (Forrest et al., 2009; Allamanis et al., 2021) finds that constraining bug creation — e.g., by restricting edits to repository history, deletions, or mutation operators — can help preserve realism. However, these approaches do not address the fully open-ended generation setting we consider here. Moreover, the constraints they impose limit the diversity and scale of synthetic training data.

We propose ANCHORED SELF-PLAY (ASP) (Figure 1) to mitigate this drift by anchoring self-play to a small reference set sampled from the target bug sources. ANCHORED SELF-PLAY (ASP) adds a code-embedding similarity re-

[1]Department of Computer Science, Stanford University, Stanford, CA, USA. Correspondence to: Caroline Choi <cchoi1@stanford.edu>.

*Proceedings of the 43$^{rd}$ International Conference on Machine Learning*, Seoul, South Korea. PMLR 306, 2026. Copyright 2026 by the author(s).

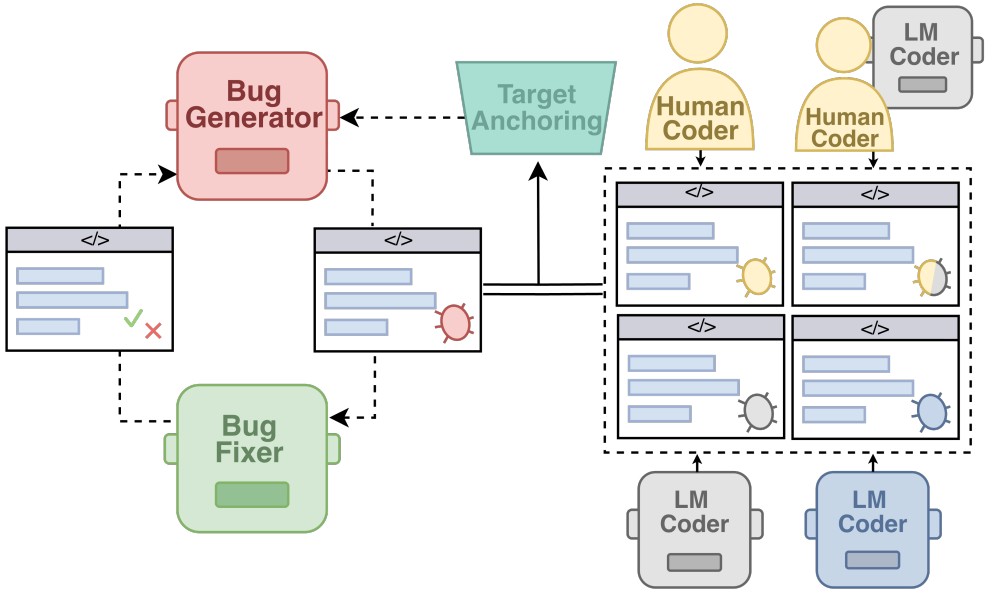

### Anchored Self-Play

### Reference Bug Sources

*Figure 1.* **Anchored self-play for code repair.** *Left:* In generator–fixer self-play, the generator edits a correct program to produce a bug and the fixer repairs it; unit tests reward bug validity and repair correctness. Because unit tests certify correctness but not realism, self-play can drift toward unrealistic yet test-failing bugs. *Right:* BUGSOURCEBENCH evaluates the same tasks under multiple realistic bug sources: human-written bugs, human edits of buggy model drafts, and naturally occurring model errors. ANCHORED SELF-PLAY (ASP) mitigates drift by anchoring training to a small reference set from these sources via an embedding-similarity reward for generation and reference mixing for fixer training.

ward to guide generation towards target-like bugs and mixes reference bugs into fixer training to prevent overfitting to the generator's evolving distribution.

In deployment, repair models encounter bugs from multiple sources – from human-authored code to LM-generated code to hybrid human-edited LM code (Cui et al., 2024). To evaluate performance across these realistic settings, we introduce BUGSOURCEBENCH, a code repair benchmark spanning three settings in LM-assisted programming: (i) human-written bugs, (ii) human-edited buggy LM code, and (iii) incorrect code produced by both weaker and stronger code LMs.

On BUGSOURCEBENCH, standard self-play improves on LM-sourced bugs but regresses on human-authored bugs, consistent with distribution drift. Anchoring reverses this failure mode: ASP achieves the best fix rates, improving average fix rate by $25\%$ (relative) / $+7.2$ pp (absolute) over standard self-play, with gains on bugs sourced from LMs ($100\%$ relative / $+11$ pp absolute) and humans ($7.1\%$ relative / $+3.4$ pp absolute).

Our key contributions are:

- We formulate open-ended generator–fixer self-play for

code repair and show it can drift, improving on LM-generated bugs while degrading on human-authored bugs.
- We propose ANCHORED SELF-PLAY (ASP), which guides bug generation with an embedding-similarity reward and reference-mixed fixer training to reduce distribution drift.
- We introduce and open-source BUGSOURCEBENCH, a controlled multi-source benchmark spanning human-written, human-edited LM, and LM-generated bugs.

## 2. Problem Formulation

Our goal is to scale supervision for training code-repair models using unit tests as the sole correctness signal. We formalize the code-repair setting and our evaluation across realistic bug sources.

**Code repair.** Each task $x$ consists of natural-language programming instructions (with input/output specifications and constraints) together with unit tests that check correctness. Given any program $c$, executing the tests produces (i) a binary verifier $v(x, c) \in \{0, 1\}$, where $v(x, c) = 1$ if and only if $c$ passes all tests for $x$, and (ii) unit test output $o(x, c)$ (e.g., compile errors, failed tests, and stack traces).

A *bug* for task $x$ is a program $b$ that (i) compiles and (ii) fails at least one test: $v(x, b) = 0$. A repair model $\pi_\theta$ (the *fixer*) induces a conditional distribution over candidate repairs given the task, buggy program, and unit test output:

$$y \sim \pi_\theta(\cdot \mid x, b, o(x, b)).$$

A repair $y$ succeeds on $(x, b)$ if it passes all tests, i.e., $v(x, y) = 1$. In our pipeline, the fixer outputs a full corrected program, rather than a diff, because producing well-formed diffs is challenging for the `Qwen2.5-Coder-7B-Instruct` model used in our experiments.

**Evaluation across bug sources.** In practice, bugs arise from heterogeneous sources, such as human programmers, LM coding assistants, or a hybrid of the two. For controlled comparisons, we evaluate on a fixed set of tasks $x$, and construct each split by sampling bugs from a different source $s$. Formally, we model each source $s \in \mathcal{S}$ as a conditional distribution over bugs for the same tasks, $b \sim P_s(\cdot \mid x)$. For a bug source $s$, the repair rate of a fixer $\pi_F$ is

$$\mathrm{Perf}(\pi_F; s) = \mathbb{E}_x \, \mathbb{E}_{b \sim P_s(\cdot \mid x)} \, \mathbb{E}_{y \sim \pi_\theta(\cdot \mid x, b, o(x, b))} \big[ v(x, y) \big].$$

Our goal is to maximize the average repair rate across bug sources,

$$\mathrm{Perf}_{\mathrm{avg}}(\pi_\theta) = \frac{1}{|\mathcal{S}|} \sum_{s \in \mathcal{S}} \mathrm{Perf}(\pi_\theta; s).$$

With this evaluation criterion in place, we next describe BUGSOURCEBENCH, which instantiates the bug source set $\mathcal{S}$ with realistic bug sources that arise in LM-assisted programming.

## 3. BUGSOURCEBENCH: Controlled Bug-Source Evaluation

### 3.1. Benchmark construction

We build BUGSOURCEBENCH from code generation tasks in BigCodeBench (Zhuo et al., 2024), which emphasize realistic library and API usage. Each BigCodeBench task provides (i) natural-language programming instructions, (ii) unit tests that define correctness, and (iii) a reference implementation that passes those tests. We convert each BigCodeBench problem into a repair instance by keeping the programming prompt and unit tests unchanged and attaching a buggy implementation.

**Task structure.** A BUGSOURCEBENCH example consists of programming instructions $x$, a buggy program $b$, and the accompanying unit-test verifier $v(x, \cdot)$. All buggy programs in BUGSOURCEBENCH compile but fail at least one test to

test semantic repair rather than syntax fixing. At evaluation, the model receives $(x, b)$ and unit-test feedback $o(x, b)$ (e.g., failing tests and truncated error traces) and must output a corrected program $y$ that passes all tests.

**Repair interface.** We compare several repair prompts: full-program repair, diff-based patching, and our default that includes unit-test traces. Test feedback improves fix rates, while diff repair often underperforms due to brittle formatting and patch application. We therefore use the test-trace interface in all main experiments; ablations are in Appendix A and Table 3.

**Bug sources.** BUGSOURCEBENCH contains four bug-source variants designed to reflect common failure modes in LM-assisted programming. All variants share the same underlying tasks $x$ (problem statement and unit tests); only the buggy program $b$ differs.

- **HUMAN.** Two annotators (one CS graduate student and one CS undergraduate) introduced bugs into each task's human-written reference solution. Annotators were instructed to make 1–4 localized edits that preserve executability (the program runs/compiles) while causing at least one unit test to fail. They were encouraged to use realistic developer mistakes (e.g., off-by-one errors, wrong constants, missing edge cases, or API misuse) rather than syntax-breaking changes.
- **HUMAN-EDITED LM.** To model human-in-the-loop revision errors, we first prompt an LM (`gpt-5-mini`) *to solve the task* and retain a sampled program that runs/compiles but fails at least one unit test. Annotators then edit this draft (e.g., adjusting indices, conditions, or initializations) while keeping it executable and still incorrect, mimicking mistakes that arise when developers modify or integrate LM-generated code.
- **LM ERRORS (QWEN-7B).** Incorrect programs produced by `Qwen2.5-Coder-7B-Instruct` when prompted *to generate correct code that completes the task* (not to generate bugs), filtered to those that run/compile but fail at least one unit test.
- **LM ERRORS (GPT-OSS-20B).** Incorrect programs produced by `gpt-oss-20b` under the same protocol, capturing subtler errors from a stronger reasoning model.

We provide per-source examples in Appendix A.1 (Figure 6) and full construction details, including the filtering criteria and sampling budgets, in Appendix A.2. Appendix A.4 presents frontier model performance on BUGSOURCEBENCH (Table 3) and distinguishes repair as a distinct skill from code generation: models that solve a task from scratch often fail to repair it, and vice versa (Table 4).

We analyze bugs from different sources in Appendix A.4. We categorize bugs into coarse error types and find systematic, source-dependent failure patterns. We then embed bugs with `voyage-code-3` and measure $k$NN source purity (excluding same-task neighbors), revealing strong within-source clustering.

With BUGSOURCEBENCH, we can test whether training on synthetic bug–fix data improves repair on the realistic bug sources in Section 2. We next present two training approaches: generator–fixer self-play with a correctness-only reward, and ANCHORED SELF-PLAY (ASP), which anchors self-play to a small reference set to better match target bug sources.

# 4. Self-Play for Code Repair

We aim to scale supervision for code repair when unit tests are the only source of verification. To do so, we propose *generator–fixer self-play*: a single policy $\pi_\theta$ is trained to synthesize and repair buggy programs of increasing difficulty.

## 4.1. Generator–fixer self-play

We train a single policy $\pi_\theta$ to play two roles: a *generator $G$* that proposes bugs and a *fixer $F$* that repairs them (Figure 1). For each task $x$ (code generation prompt, tests, and a reference solution), the generator samples a candidate buggy program

$$b \sim \pi_\theta(\cdot \mid x),$$

and we run unit tests to obtain test output $\mathrm{o}(x, b)$. The fixer then samples one or more candidate repairs conditioned on the task, buggy program, and test output,

$$y \sim \pi_\theta(\cdot \mid x, b, \mathrm{o}(x, b)).$$

We say $b$ is *valid* if it compiles and fails at least one unit test; invalid bugs receive a penalty and are not passed to the fixer.

## 4.2. Correctness and difficulty shaping

**Fixer reward.** The fixer is rewarded for producing correct repairs. Concretely, for a repair $y$ we use

$$r^{\mathrm{F}}(x, b, y) = v(x, y),$$

which is 1 if and only if $y$ passes all tests and 0 otherwise.

**Generator reward.** As the fixer improves, the generator should produce progressively harder but still solvable bugs to continue improving the fixer's repair capability.

A single repair attempt is noisy, so we estimate bug difficulty using $K$ independent repair attempts. For a bug $(x, b)$, we

sample $K$ repairs $y^{(1)}, \ldots, y^{(K)} \sim \pi_\theta(\cdot \mid x, b, \mathrm{o}(x, b); \mathrm{FIX})$ and define the fix rate

$$\rho(x, b) = \frac{1}{K} \sum_{k=1}^{K} v(x, y^{(k)}).$$

However, the generator can collapse to invalid bugs (e.g. compile or syntax errors) or bugs that are unsolvable by the current fixer, providing little training signal. We shape generator rewards using $\rho(x, b)$, rewarding valid bugs that fall in a moderate difficulty band:

$$r^{G}_{\mathrm{base}}(x, b) = \begin{cases} -1, & b \text{ does not compile or passes all tests,} \\ 1, & \rho(x, b) \in [\rho_\ell, \rho_h], \\ -\alpha, & \rho(x, b) \in \{0, 1\}, \\ 0, & \text{otherwise.} \end{cases}$$

## 4.3. Optimization

We optimize the generator-fixer self-play loop with GRPO. For each task $x$, we first sample $G{=}4$ candidate bugs $b_i \sim \pi_G(\cdot \mid x)$. For each bug $(x, b_i)$, we then sample $K{=}4$ independent repair attempts $y_i^{(1:K)} \sim \pi_F(\cdot \mid x, b_i, \mathrm{o}(x, b_i))$ and compute the fix rate $\rho(x, b_i) = \frac{1}{K} \sum_{k=1}^{K} v(x, y_i^{(k)})$.

We write the generator reward for proposing $b$ on task $x$ as $R^{\mathrm{G}}(x, b)$ and the fixer reward for producing repair $y$ on $(x, b)$ as $R^{\mathrm{F}}(x, b, y)$. In standard self-play, $R^{\mathrm{F}}$ is the unit-test pass indicator and $R^{\mathrm{G}}$ is solve-rate–shaped (Section 4.2); ASP adds auxiliary terms to $R^{\mathrm{G}}$ and occasionally replaces $b$ with $b^{\mathrm{ref}}$ when updating the fixer (Section 4.5).

**Generator update.** We update the generator with Group Relative Policy Optimization (GRPO). For each task $x$, we sample $G$ candidate bugs $\{b_i\}_{i=1}^{G}$ and compute group-normalized advantages:

$$\hat{A}_i^G = \frac{R^{\mathrm{G}}(x, b_i) - \mu^G(x)}{\sigma^G(x) + \epsilon},$$

$\mu^G(x), \sigma^G(x)$ computed over $i \in \{1, \ldots, G\}$.

We then optimize a GRPO objective using a clipped policy-gradient update on $\pi_\theta(\cdot \mid x)$.

**Fixer update.** We update the fixer with Group Relative Policy Optimization (GRPO). Using the same sampled bugs, we assign each repair attempt the reward $R^{\mathrm{F}}(x, b_i, y_i^{(k)})$. For each bug $b_i$, we compute a per-bug baseline across the $K$ repairs and form group-normalized advantages:

$$\hat{A}_{i,k}^F = \frac{R^{\mathrm{F}}(x, b_i, y_i^{(k)}) - \mu^F(x, b_i)}{\sigma^F(x, b_i) + \epsilon},$$

$\mu^F(x, b_i), \sigma^F(x, b_i)$ computed over $k \in \{1, \ldots, K\}$.

We then optimize a GRPO objective using a clipped policy-gradient update on $\pi_\theta(\cdot \mid x, b, o(x, b))$, computing the loss only on tokens generated by the policy and masking prompt tokens.

### 4.4. Distribution drift under standard self-play

The base rewards above ensure bugs are valid and appropriately difficult, but they do not encourage *realistic* bug patterns. As a result, the self-play distribution can drift toward idiosyncratic edits that remain valid and informative for training, yet differ from bugs written by humans or produced by LM coding assistants.

In Figure 2b, we observe that standard self-play shows small initial gains but performance later degrades on Human-sourced bugs despite the fixer improving on its self-generated bugs in Figure 2a. We attribute this to distribution drift: the generator produces increasingly synthetic bug patterns that the fixer overfits to. We thus modify the training objectives in self-play to anchor bug generation to reference bug sources.

### 4.5. Anchoring self-play to reference bug sources

We assume access to a small dataset of reference-source bugs, $\mathcal{D}_{\mathrm{ref}}$, drawn from the training tasks and disjoint from evaluation. We use $\mathcal{D}_{\mathrm{ref}}$ in two ways: (i) *reference mixing*, which injects reference bugs into fixer training, and (ii) *similarity-guided shaping*, which nudges the generator toward reference-like bugs.

#### 4.5.1. REFERENCE MIXING FOR FIXER TRAINING

For tasks with an associated reference bug $b^{\mathrm{ref}} \in \mathcal{D}_{\mathrm{ref}}$, we replace the generated bug with probability $p_{\mathrm{mix}}$ and train the fixer on $(x, b^{\mathrm{ref}}, o(x, b^{\mathrm{ref}}))$. On mixed episodes, we do not update the generator to preserve on-policy generation dynamics; empirically, updating the generator on these mixed episodes reduced performance (Table 10).

#### 4.5.2. SIMILARITY-GUIDED GENERATOR SHAPING

Reference mixing (above) trains the fixer but does not encourage the generator to produce reference-like bugs. We therefore add an auxiliary reward to the generator based on similarity to bugs in $\mathcal{D}_{\mathrm{ref}}$.

**Edit embedding and kNN similarity.** For each generated bug $b$, we compute a unified diff between the reference solution and $b$, embed the diff with a code embedding model (`voyage-code-3`), and compute its average cosine similarity to the $k$ nearest neighbors in a pool of reference edit embeddings. We map the resulting score to $[0, 1]$ and denote it by $\mathrm{sim}_{01}(b)$.

**Baseline subtraction.** The code-embedding similarity term should act as a *shaping* signal, biasing generation toward reference-like bugs, without overwhelming the base unit-test reward. Rather than relying on a fixed scale (which can be sensitive to the similarity model and batch composition), we center the similarity scores with a running baseline:

$$\delta_t(b) = \mathrm{sim}_{01}(b) - B_t, \quad B_t \leftarrow \beta B_{t-1} + (1-\beta)\,\mathbb{E}[\mathrm{sim}_{01}(b)],$$

where the expectation is taken over the current batch. This makes $\delta_t$ measure *relative* similarity – how much more (or less) reference-like a candidate is than what the generator currently produces on average. For valid, generated (non-mixed) bugs, the final generator reward is

$$r^{\mathrm{G}}(x, b) = r^{\mathrm{G}}_{\mathrm{base}}(x, b) + \lambda \operatorname{clip}(\delta_t(b)).$$

We found this centering more stable than tuning a single global scale, since it is invariant to shifts in the absolute magnitude of similarity scores. Hyperparameters are in Appendix B.1.

## 5. Experimental Setup

**Data and splits.** We train on 900 BigCodeBench tasks (Zhuo et al., 2024). For testing, we use 127 held-out tasks that appear in all BUGSOURCEBENCH splits; thus the underlying tasks are shared and only the buggy programs differ across sources. For validation and checkpoint selection, we use 81 tasks from all BUGSOURCEBENCH splits, disjoint from train and test. We report results on BUG-SOURCEBENCH-HUMAN, BUGSOURCEBENCH-HUMAN-EDITED LM, BUGSOURCEBENCH-QWEN-7B, and BUG-SOURCEBENCH-GPT-OSS-20B.

**Reference pool (anchoring data).** ASP maintains a reference pool of 900 bugs sampled from the training splits of each bug source in BUGSOURCEBENCH, with an equal number drawn from each of the four sources. This pool is used both (i) to construct the reference embedding pool for the similarity reward and (ii) as reference-mixed data when training the fixer. For parity, Fixer-Only also mixes the same reference bugs into fixer training, but does not use our similarity-based anchoring.

**Initialization.** Both generator and fixer are initialized from `Qwen2.5-Coder-7B-Instruct` (Hui et al., 2024). Unless stated otherwise, a single set of weights is shared between roles and trained with GRPO.

**Comparisons.** We compare ASP with:

- **Base Model:** pretrained model used as a fixer without training.

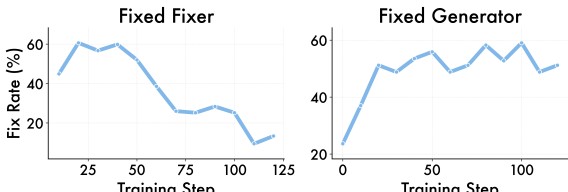 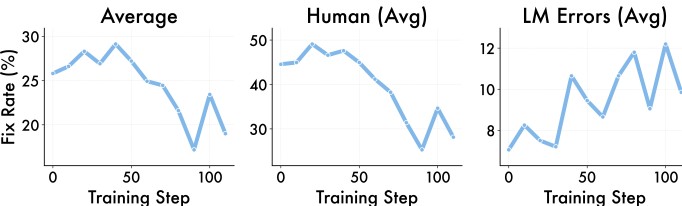

*(a)* **Co-evolution of standard self-play.** With a fixed fixer checkpoint (step 40), fix rate declines over generator training, indicating the generator produces harder bugs. With a fixed generator checkpoint (step 40), fix rate increases as the fixer trains, reflecting improved repair ability on the generated bugs.

*(b)* **Standard self-play exhibits distribution drift.** Fix rate improves early but later regresses, most strongly on human-originated splits. This suggests overfitting to the generator's shifting synthetic bug distribution.

*Figure 2.* **Self-play training dynamics and distribution drift.**

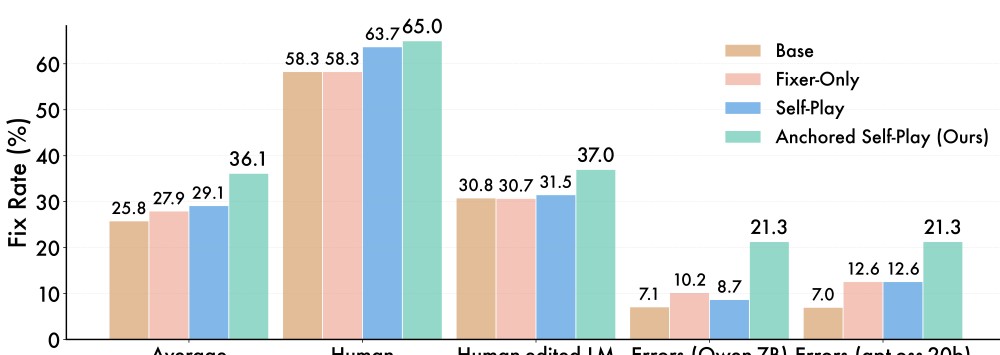

*Figure 3.* **Fix rates across bug sources on BUGSOURCEBENCH.** Standard self-play improves on LM-sourced bugs but regresses on human-authored bugs, consistent with distribution drift; anchoring (ASP) reverses this failure mode and achieves the best overall fix rate (+7.0 pp / +24% relative vs. self-play), with gains on both LM (+11 pp / +100% rel.) and human bugs (+3.4 pp / +7.1% rel.).

- **Fixer-Only (frozen generator):** freeze the generator at the base model and train only the fixer on bugs sampled from this static generator; reference bugs are mixed into fixer batches for parity with ASP.
- **Self-Play (joint training):** alternate between bug generation and repair, updating both roles with GRPO; weights are shared.

**Training and evaluation protocol.** For each task, the generator samples candidate buggy programs; we keep only programs that execute and fail at least one unit test. The fixer is prompted with the problem and buggy code, and optionally a truncated summary of failing unit-test output. During training, the fixer attempts up to $K$ repairs per bug (default $K{=}4$) and receives unit-test feedback as the reward. We update policies with GRPO using role-normalized advantages. At test time, we use a single repair attempt per bug and decode greedily (temperature 0.0). Full hyperparameters and prompts are provided in Appendix B. Additional baselines are compared in Appendix E.

## 6. Main Results

We report main results comparing ASP, standard self-play, and fixer-only training in Figure 3. ASP achieves the strongest performance across all bug sources, improving overall fix rate by 24% (relative) / +7.0 pp (absolute) over standard self-play and 29% (relative) / +8.2 pp (absolute) over fixer-only training. The largest gains are on bugs sourced from LMs: +145% (relative) / +12.6 pp (absolute) on Qwen-7B, and +69% (relative) / +8.7 pp (absolute) on gpt-oss-20b. Importantly, anchoring also improves performance on human-written bugs relative to standard self-play: +17.5% (relative) / +5.5 pp (absolute) on Human-Edited and +2.0% (relative) / +1.3 pp (absolute) on Human.

Standard self-play improves the overall average over fixer-only (+1.2 pp), but does *not* consistently help across sources: it degrades on Qwen-7B (−1.5 pp). In contrast, ASP improves *every* source simultaneously. The base model and Fixer-Only tie on average performance, yet fixer-only training substantially improves LM subsets (Qwen-7B: 10.2% vs. 7.1%; gpt-oss-20b: 12.6% vs. 7.0%) with negligible change on Human (30.7% vs. 30.8%). This underscores the importance of evaluating repair across bug sources. We provide further evaluations of pass@$k$ performance in Appendix D.1. ASP can also improve larger 30B+ coders when used as a test-time fixer (Appendix D.5).

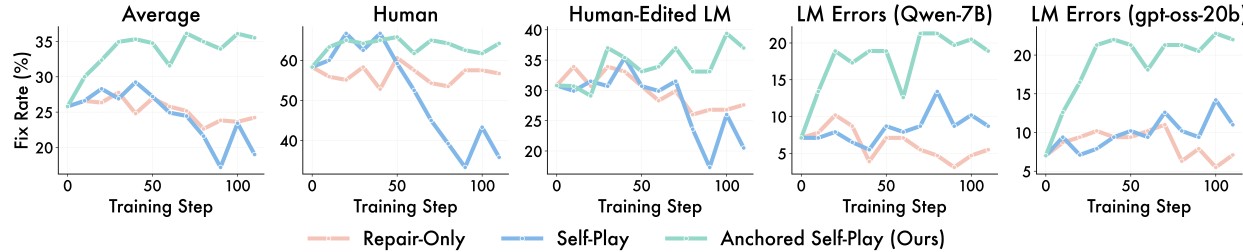

*Figure 4.* **Anchoring stabilizes self-play and improves cross-source repair.** Fix rate (%) vs. training step for Fixer-Only, vanilla Self-Play, and ASP (ours), evaluated on held-out bugs from multiple sources (gpt-oss-20b, Human, Qwen-7B, and Human-Edited LM) and their average. Standard self-play shows early gains but later degrades on realistic sources (notably Human and Human-Edited LM), consistent with distribution drift, while anchored self-play yields higher and more stable performance across sources.

**ASP produces more reference-like bugs.** We validate that the generator's similarity reward shifts the bug distribution using two embedding-based diagnostics in Figure 9. First, the mean kNN similarity of generated bugs to the reference pool increases over training under ASP, indicating that shaping moves generations toward the reference source under the same signal used for optimization. Second, when we stratify reference-benchmark performance by these similarity quantiles, ASP outperforms standard self-play within each bin, suggesting that the gains are not explained solely by producing higher-similarity bugs (Appendix D.3). Evaluating across BUGSOURCEBENCH sources by semantic bug categories shows that ASP improves consistently across bug types (Appendix D.4).

Finally, we provide qualitative examples of generated bugs before and after training with ASP in Figure 8 (Appendix D).

## 7. Ablations

### 7.1. Anchoring components

Table 1a ablates the two components of ASP: (i) mixing a small set of reference bugs into training (*Ref. Mix*) and (ii) adding an embedding-based code-similarity reward for the generator (*Sim. Reward*). Starting from Self-Play, each component alone provides a small improvement in fix rate. Combining them yields larger improvements because they address complementary failure modes in the generator–fixer loop. The similarity reward guides the generator toward reference-like bugs, while reference mixing stabilizes the fixer by providing exposure to realistic bugs as the generator evolves.

### 7.2. Reference set sources

Table 1b varies the bug sources included in the reference set. Using only HUMAN and HUMAN-EDITED LM references yields the best performance on human bugs, suggesting that anchoring on human patterns preserves robustness on human sources. Using only QWEN-7B and GPT-OSS-20B refer-

*(a)* **Ablation of components in ASP.** "Ref. mix" mixes a small set of reference bugs into training, while "Sim. reward" adds an embedding-based code-similarity reward to guide the generator. Combining both yields the best fix rate. Results are averaged over all BUGSOURCEBENCH splits.

| Method | Ref. Mix | Sim. Reward | Fix (%) |
|---|---|---|---|
| Base Model | | | 25.8 |
| Self-Play | | | 29.1 |
| + Reference Bug Mixing | ✓ | | 29.5 |
| + Similarity Shaping | | ✓ | 30.9 |
| **ASP** (Ours) | ✓ | ✓ | **36.1** |

*(b)* **Effect of reference pool composition.** We vary the reference set used for anchoring (human-only, LM-only, or mixed) and report fix rates on each BUGSOURCEBENCH split. Human-only references yield the best overall performance and strongest gains on human and human-edited bugs, while LM-only references shift improvements toward LM-originated bugs. This illustrates how the reference set steers which bug patterns self-play emphasizes.

| Ref. Pool | Fix Rate (%) | | | | |
|---|---|---|---|---|---|
| | Overall | Human | Hum.-Ed. | Qwen | GPT-OSS |
| Human-only | 34.4 | 67.5 | 37.8 | 11.8 | 11.0 |
| LM-only | 32.0 | 65.4 | 41.7 | 13.4 | 17.3 |
| Self-Play | 29.1 | 63.7 | 31.5 | 8.7 | 12.6 |
| **ASP** (Ours) | **36.1** | 65.0 | 37.0 | 21.3 | 21.3 |

ences improves repair on LM-originated bugs but slightly degrades human performance. The reference set guides learning toward the bug patterns it contains. These results motivate using a reference set that contains multiple bug sources.

### 7.3. Reference set size

Figure 5 varies the reference set size, sampled uniformly across bug sources: Human, Human-Edited LM, Qwen-7B, and gpt-oss-20b. ASP improves with as few as 50 reference examples, indicating sample-efficient anchoring. Repair performance improves with larger reference sets, which provide broader coverage of reference bug patterns.

Additional robustness ablations in Appendix D show that ASP's gains and drift are consistent across (i) the embedding model and kNN pooling parameter $k$ (Table 9), (ii) different base models (Table 12), (iii) (de)coupling of generator/-fixer weights (Table 11), and (iv) different task distributions (Table 13).

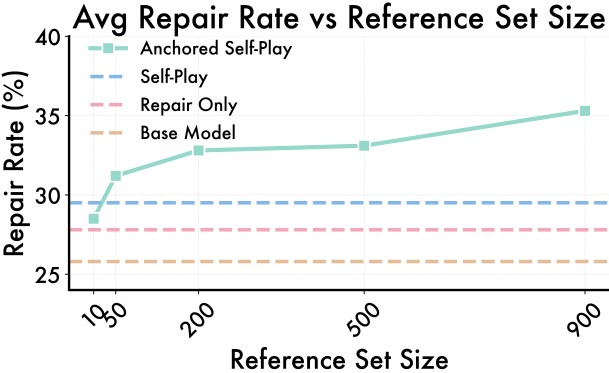

*Figure 5.* **Reference set scaling.** Fix rate of ASP as a function of the number of reference bugs used for anchoring (x-axis). We report performance on each BUGSOURCEBENCH split and the overall average. Overall repair improves steadily as the reference set grows, with the largest gains on LM-generated bug sources, while human and human-edited splits remain strong across sizes.

## 8. Related Work

**Code repair and bug-source variation.** Program repair is commonly evaluated on curated real-world bug datasets from open-source projects (e.g., Defects4J, BugsInPy, ManyBugs, Bears) and on short, unit-testable repair benchmarks (e.g., QuixBugs) (Just et al., 2014; Widyasari et al., 2020; Le Goues et al., 2015; Madeiral et al., 2019; Lin et al., 2017). In parallel, test-driven code generation benchmarks (e.g., HumanEval, MBPP, APPS, EvalPlus, Live-CodeBench) highlight that fluent generations often fail functional correctness (Chen et al., 2021; Austin et al., 2021; Hendrycks et al., 2021; Liu et al., 2023; Jain et al., 2024). More recently, repository-level benchmarks and software-engineering agents emphasize end-to-end issue resolution with realistic context and tooling (e.g., SWE-bench and follow-ons, SWE-agent, Agentless, PatchPilot, Co-PatcheR, and OpenHands) (Jimenez et al., 2023; Yang et al., 2025a; Pham et al., 2025; Yang et al., 2024; Xia et al., 2024; Li et al., 2025; Tang et al., 2025; Wang et al., 2025a). Across settings, a recurring theme is sensitivity to the bug distribution and data-generating process, with substantial shifts between human-written bugs, synthetic mutations, and errors in model-generated code (He et al., 2022; Xu et al., 2022; Sonwane et al., 2025; Dou et al., 2025; Yang et al., 2025b). BUGSOURCEBENCH complements prior benchmarks by holding task format fixed (short, unit-testable problems with real library/API usage) while explicitly constructing splits from distinct bug sources, enabling controlled measurement of cross-source generalization and self-play drift.

**Learning from synthetic bugs under grounding or constrained transformations.** Several approaches obtain supervision by generating synthetic bugs but constrain the bug space to reduce pathological drift. BugLab uses predefined mutation operators to generate defects for self-supervised localization/repair (Allamanis et al., 2021; Forrest et al., 2009). In repository settings, recent data-generation and self-play systems leverage external grounding (e.g., repository structure, tests, and patch provenance) to keep training tasks realistic at scale (Yang et al., 2025a; Pham et al., 2025; Wei et al., 2025; Ye et al., 2023; Zirak & Hemmati, 2024). Close to our setting is Break-it-Fix-it (BIFI), which trains paired bug introducers and fixers using a critic (e.g., a parser/compiler) and biases the introducer toward more natural corruptions (Yasunaga & Liang, 2021; Long & Rinard, 2016; Chen et al., 2019). In contrast, we study fully synthetic short-form self-play where the generator can apply arbitrary text edits and unit tests certify failure but only weakly constrain realism; we therefore shape generation with an embedding-similarity reward that remains compatible with open-ended bug generation.

**Self-play and automatic curriculum generation.** A growing line of work uses self-play to generate curricula by proposing tasks near a learner's frontier and learning from rollouts (Bengio et al., 2009; Silver et al., 2017; Cheng et al., 2024; Kuba et al., 2025; Chen et al., 2024; Zhao et al., 2025; Huang et al., 2025). In open-ended text settings, self-play has been used to synthesize training distributions for reasoning and theorem proving (Poesia et al., 2024; Dong & Ma, 2025; Chen et al., 2025; Liu et al., 2025; Yu et al., 2025). In language and program synthesis, self-generated curricula have also been studied through adaptive testing and open-ended/autotelic task design (Ribeiro & Lundberg, 2022; Colas et al., 2022; Parker-Holder et al., 2023; Teodorescu et al., 2023; Pourcel et al., 2024). For code specifically, proposer–solver variants couple program synthesis with test generation or formal verification (Lin et al., 2025; Wang et al., 2025b; Wilf et al., 2025). Our work adapts the curriculum perspective to code repair, but focuses on a core failure mode in the unit-test setting: without an explicit realism signal, self-play can drift toward test-breaking edits that are difficult yet unrepresentative of bugs encountered in practice. We mitigate this by mixing reference bugs into fixer training and adding an embedding-based similarity reward for the generator.

## 9. Discussion

We study whether unit tests alone can scale supervision for code repair via open-ended synthetic data generation. We introduce generator–fixer self-play, which trains a single model with reinforcement learning to alternate between generating bugs (tests fail) and repairing them (tests pass), and find that this setting can drift toward valid yet unrealistic bugs, improving in-distribution repair while degrading on human-written bugs. To mitigate drift without repository grounding, we propose ANCHORED SELF-PLAY (ASP), which anchors self-play with a small reference set via an

embedding-similarity reward for bug generation and reference mixing for fixer training, and evaluate bug-source generalization using BUGSOURCEBENCH, a controlled benchmark spanning human-written bugs, human-edited bugs in model drafts, and bugs from model-generated code. Overall, ASP improves average repair over standard self-play and narrows the gap between model-generated and human bug sources, highlighting the need for explicit realism signals in unit-test-only self-play. Directions for future work include stronger realism objectives (e.g., learned or preference-based bug-style critics) and richer anchoring signals beyond embeddings.

**Limitations.** Our experiments focus on Python, function-level repair tasks with unit-test feedback. This controlled setting lets us isolate bug-source shift and self-play drift, but it does not capture repository-level challenges such as fault localization across many files, dependency management, build-system interaction, or long-horizon tool use. Extending ASP to SWE-bench-style agentic repair would require substantially larger models and infrastructure, and remains an important direction. In addition, our realism signal is based on a frozen code-embedding similarity proxy and a finite reference pool; while our ablations suggest robustness to the embedding model and nearest-neighbor parameter, learned critics or human preference data may provide stronger realism signals in future work.

## Impact Statement

This work studies LMs for bug fixing, with the goal of improving the reliability of AI-assisted software development. By analyzing how bug-fixing systems generalize across different sources of errors, our work can contribute to more reliable automated programming tools. While this may increase reliance on automated systems, we do not anticipate negative societal impacts beyond those already associated with the broader use of machine learning in software engineering.

## Software and Data

Code is available at

github.com/cchoi1/anchored_self_play.

## Acknowledgements

We thank Yangjun Ruan, Neil Band, Kaiyue Wen, Luke Bailey, Thomas Chen, and Arvind Mahankali for helpful discussions and feedback on the paper draft. We also thank the anonymous reviewers for their helpful feedback. CC was supported by the National Science Foundation Graduate Research Fellowship Program under Grant No. DGE-2146755.

# A. BUGSOURCEBENCH

## A.1. Examples

---

**BugSourceBench TaskID 1063**

Performs Principal Component Analysis (PCA) on the sum of rows of a 2D numpy array and plots the explained variance ratio.
*Note that:* the title of the plot is set to "Explained Variance Ratio of Principal Components". The function assumes that 'arr' is a valid 2D numpy array. Only the first principal component is considered in this analysis. The plot illustrates the proportion of the dataset's variance that lies along the axis of this first principal component.
*The function should output with:* ax (matplotlib.axes.Axes) – an Axes object from matplotlib.

---

**Human**

```python
from matplotlib import pyplot as plt
from sklearn.decomposition import PCA
def task_func(arr):

    row_sums = arr.sum(axis=1)

    pca = PCA(n_components=10)
    pca.fit(row_sums.reshape(-1, 1))

    # Plotting (requires matplotlib and sklearn)

    _, ax = plt.subplots()

    ax.bar([0], pca.explained_variance_ratio)
    ax.set_title("Explained Variance Ratio of Principal
        Components")
    ax.set_xticks([0])
    ax.set_xticklabels(["PC1"])

    return ax
```

**Human-Edited LM**

```python
from matplotlib import pyplot as plt
from sklearn.decomposition import PCA
def task_func(arr):
    if arr.size == 0:
        raise ValueError("Input array is empty.")

    summed = np.sum(arr, axis=1)
    X = summed.reshape(-1, 1)
    pca = PCA(n_components=1)
    pca.fit(X)

    fig, ax = plt.subplots()
    ax.bar([1], pca.explained_variance_, color="tab:blue")
    ax.set_xticks([1])
    ax.set_xticklabels(["PC1"])
    ax.set_ylabel("Explained Variance Ratio")
    ax.set_title("Explained Variance Ratio of Principal
        Components")

    return ax
```

**LM Errors (Qwen-7B)**

```python
from matplotlib import pyplot as plt
from sklearn.decomposition import PCA
def task_func(arr):
    # Ensure arr is a 2D numpy array
    if len(arr.shape) != 2:
        raise ValueError("Input must be a 2D numpy array")

    # Sum the rows of the array
    summed_data = np.sum(arr, axis=0)

    # Reshape the data to be 2D for PCA
    reshaped_data = summed_data.reshape(1, -1)

    # Perform PCA
    pca = PCA(n_components=1)
    pca.fit(reshaped_data)

    # Get the explained variance ratio
    explained_variance_ratio = pca.explained_variance_ratio_

    # Create a plot
    fig, ax = plt.subplots()
    ax.bar(range(1), explained_variance_ratio,
        tick_label=['PC1'])
    ax.set_title('Explained Variance Ratio of Principal
        Components')

    return ax
```

**LM Errors (gpt-oss-20b)**

```python
from matplotlib import pyplot as plt
from sklearn.decomposition import PCA
def task_func(arr):
    # Sum over rows to get column totals
    row_sums = arr.sum(axis=0)

    # Reshape to 2D array for PCA (n_samples=1,
        n_features=number of columns)
    X = row_sums.reshape(1, -1)

    # Perform PCA with a single component
    pca = PCA(n_components=1)
    pca.fit(X)

    # Extract explained variance ratio for the first component
    evr = pca.explained_variance_ratio_[0]

    # Plot the explained variance ratio
    fig, ax = plt.subplots()
    ax.bar([1], [evr], width=0.5, color='skyblue')
    ax.set_title("Explained Variance Ratio of Principal
        Components")
    ax.set_xlabel("Principal Component")
    ax.set_ylabel("Explained Variance Ratio")
    ax.set_xticks([1])
    ax.set_ylim(0, 1)

    return ax
```

*Figure 6.* BUGSOURCEBENCH example showing buggy programs from different sources for the same task $x$. Bugs differ qualitatively across sources: the human code uses incorrect values that misalign with the spec and misuses the API, the human edit contains multiple logic mistakes (e.g., using absolute variance, forgetting imports), Qwen-7B makes off-by-one errors by using the incorrect axis and shape, and gpt-oss-20b makes similar model mistakes while also showing instruction-following issues.

## A.2. Construction

**Overview.** BUGSOURCEBENCH is built on a fixed set of 1,114 BigCodeBench-style programming tasks. Each task provides natural-language instructions and unit tests that define correctness. Every BUGSOURCEBENCH instance pairs a task with a *buggy program* that (i) runs/compiles and (ii) fails at least one unit test, so repair requires a semantic fix rather than syntax cleanup. All BUGSOURCEBENCH variants share the same underlying tasks and differ *only* in how the buggy program is produced, enabling controlled comparisons across bug sources.

**Bug validity criteria.** We first remove 26 tasks from the BigCodeBench dataset whose reference (ground-truth) solutions do not pass the unit tests, leaving $1, 114$ tasks in which to generate bugs. We accept a candidate program as a bug if it compiles successfully but fails at least one unit test. Each generated bug is validated by executing the task's unit tests using our standard reward harness. We accept a candidate if (i) it is *not* correct (fails at least one unit test) and (ii) it does

*not* trigger compilation or runtime failures (identified via robust pattern matching over test output, e.g., `SyntaxError`, `ImportError`, `NameError`; assertion failures are treated as valid test failures).

**Task structure.**    All datasets are converted to a consistent BUGSOURCEBENCH-compatible schema. We store *function bodies only* (4-space indented) for both `buggy` and `canonical_solution`. Each example contains:

- `task_id`: Unique identifier for the underlying programming task (shared across bug sources).
- `instruct_prompt`: Natural-language problem statement, including input/output specifications and constraints.
- `buggy`: Buggy solution code for this task (function body only) that runs but fails at least one unit test.
- `canonical_solution`: Reference correct solution code (function body only) used as the ground-truth implementation. *Note: this is never provided to the fixer.*
- `test`: Unit-test harness used to evaluate candidate solutions (typically includes the test cases and a runner).
- `complete_prompt`: Full text prompt given to the model in our default repair interface (instructions + buggy code + test feedback, formatted for the model).
- `code_prompt`: Code-only prompt segment that contains the program context the model is expected to modify/replace (e.g., function signature + buggy body), without natural-language instructions.
- `entry_point`: Name of the function to be implemented/repaired (used by the test harness to call into the solution).
- `doc_struct`: Structured metadata extracted from the problem statement (e.g., function signature, arguments, return type, or other parsed specification fields when available).
- `libs`: List of libraries/modules permitted or required by the task (used to reproduce the intended execution environment).
- `test_output` (added by us): Truncated unit-test feedback produced by running `buggy` on `test` (e.g., failing test names and error traces), used for analysis and as repair context.

This unified schema allows swapping bug sources while holding tasks (instructions and tests) fixed.

**Bug sources.**    We construct four bug-source variants, each providing a different buggy program for the *same* underlying BigCodeBench tasks. For fair comparison, we keep only the intersection of `task_ids` that appear in *all* variants.

- **BUGSOURCEBENCH-HUMAN.** Starting from each task's human-written reference solution, two annotators (one CS graduate student and one CS undergraduate) introduce 1–4 localized edits that cause the program to fail at least one unit test.
- **BUGSOURCEBENCH-HUMAN-EDITED LM.** For each task, we prompt `gpt-5-mini` with the original BigCodeBench instructions *to solve the task* (we do not prompt it to introduce bugs), and resample up to 16 times until we obtain a program that compiles but fails at least one unit test. We retain one such draft, then have annotators edit it (e.g., changing loop indices or initializations) while keeping it syntactically valid and still incorrect (i.e., it executes and fails at least one test). Tasks where no such draft is found within the budget are removed from BUGSOURCEBENCH.
- **BUGSOURCEBENCH-QWEN-7B.** For each task, we prompt `Qwen2.5-Coder-7B-Instruct` *to solve the task* (not to generate a bug) and resample up to 16 times until we obtain a program that compiles but fails at least one unit test, retaining one such program per task.
- **BUGSOURCEBENCH-GPT-OSS-20B.** We use the same procedure with `gpt-oss-20b`: prompt it *to solve the task* and resample up to 16 times until we obtain a program that compiles but fails at least one unit test.

**Task alignment and splits.**    We begin from BUGSOURCEBENCH-style datasets that each provide buggy solutions for a common pool of BigCodeBench tasks. To ensure every bug source is evaluated on an identical task set, we compute the intersection of `task_ids` *separately for each split* across all variants, and filter each variant to that intersection. We provide two evaluation modes: a large test set (`test_all`) of 517 examples per bug source (2068 instances total) and a smaller test set (`test`) of 127 examples per bug source (508 instances total). Note that our main experiments use the smaller `test` split for evaluation. We also create a `train` split for each variant by taking task ids not in `test`/`test_all` and intersecting them with BigCodeBench-`train` to avoid leakage; we use these training splits to form the reference pools for ANCHORED SELF-PLAY (ASP).

### A.3. Bug Source Analyses

We characterize bugs with their main types and properties. In our main bug type classification, we identify 5 main categories of bugs: LOGIC_ERROR includes bugs where the algorithm or reasoning is incorrect; WRONG_VALUE includes bugs where a specific identifier, literal, or constant is wrong, such as typos, wrong returns, or off-by-one errors;

MISSING_EDGE_CASE includes improper or missing handling of edge cases or validations; API_MISUSE includes using a framework API or library improperly, such as wrong methods and types; and OTHER includes those that don't fall directly under one of the prior categories, including missing imports, syntax errors, and others.

With these guides, we label each buggy program with a coarse bug category using GPT-4o, conditioned on the task specification, reference solution, buggy code, and unit-test traces. The results, along with kNN source clustering analysis of the buggy codes, are presented in Figure 7.

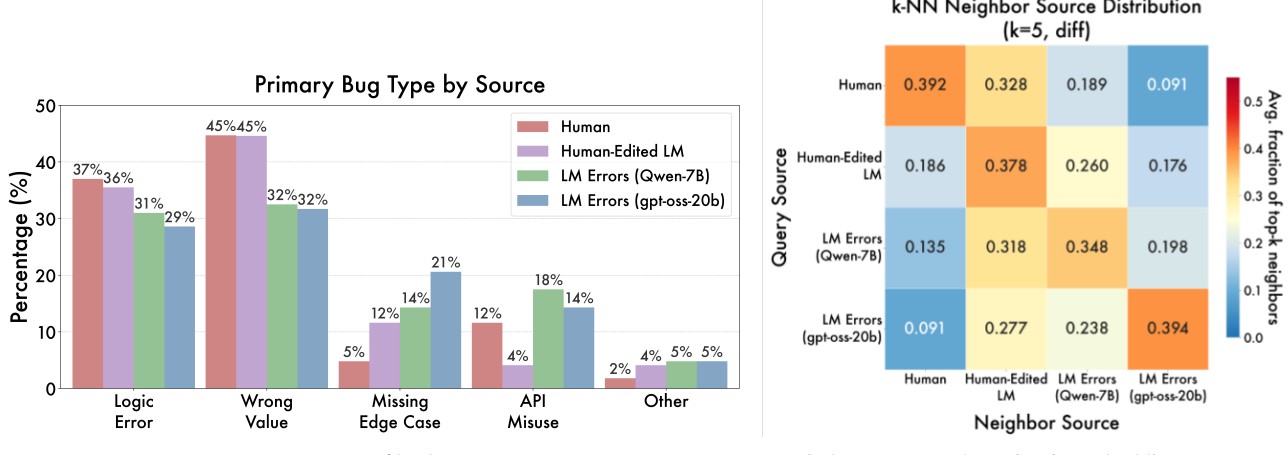

*(a)* Bug-type profiles by source.  *(b)* kNN source clustering in embedding space.

*Figure 7.* **Characterizing bug sources.** We label each buggy program with a coarse category using GPT-4o and report the resulting distribution for each BUGSOURCEBENCH split. Human edits skew toward logical errors, gpt-oss-20b toward edge-case and constraint violations, and Qwen-7B toward type/API mistakes. For each bug, we embed the diff of code from the reference with `voyage-code-3` and compute the fraction of its $k$ nearest neighbors, excluding neighbors from the same task, that come from each source. Diagonal dominance indicates within-source clustering.

*Table 2.* The test case failure rates of bugs across sources.

| Source | Mean Fail Rate | 100% Fail |
|---|---|---|
| Human | 66.2% | 31.5% |
| Human-Edited LM | 54.2% | 17.3% |
| LM Errors (Qwen-7B) | 64.5% | 29.1% |
| LM Errors (gpt-oss-20b) | 53.5% | 20.5% |

We further analyze the failure modes of bugs from various sources. We compare the proportion of failed tests of bugs from each source in Table 2. Bugs from the HUMAN source have the highest fail rate and proportion of bugs that fail all unit tests, while bugs from the HUMAN-EDITED LM source are often more subtle and pass multiple unit tests.

### A.4. Evaluation of Frontier Models on BUGSOURCEBENCH

Table 3 reports fix rates for several frontier models across bug sources and repair interfaces. We summarize several key findings.

*Table 3.* **Repair interface comparison.** Fix rate (%) across BUGSOURCEBENCH bug sources for REPAIR (full repair), +TESTS (repair with unit-test traces), and DIFF (patch output). CODEGEN is accuracy on the original code-generation tasks.

| Model | Codegen Score | Human Repair | Human +Tests | Human Diff | Human-Edited LM Repair | Human-Edited LM +Tests | Human-Edited LM Diff | LM (Qwen-7B) Repair | LM (Qwen-7B) +Tests | LM (Qwen-7B) Diff | LM (gpt-oss-20b) Repair | LM (gpt-oss-20b) +Tests | LM (gpt-oss-20b) Diff |
|---|---|---|---|---|---|---|---|---|---|---|---|---|---|
| **GPT-5.2** | 46.5 | 67.7 | **67.7** | 62.2 | 36.2 | **53.5** | 32.3 | 19.7 | **44.9** | 18.1 | 17.3 | **50.4** | 18.9 |
| **o4-mini** | 48.0 | 66.1 | **70.9** | 55.1 | 37.8 | **56.7** | 31.5 | 18.9 | **44.9** | 9.4 | 15.0 | **49.6** | 9.4 |
| **Sonnet** | 48.0 | 76.4 | **81.1** | 72.4 | 39.4 | **51.2** | 35.4 | 15.7 | **44.9** | 14.2 | 13.4 | **44.9** | 14.2 |

**Bug-source shift significantly affects performance.** Performance varies substantially with bug source. Bugs from incorrect model-generated drafts (Qwen-7B, gpt-oss-20b) are consistently harder than human or human-edited bugs.

## A.5. Ablation of Repair Interfaces for BUGSOURCEBENCH

We compared experiments of models with several repair interfaces commonly used in practice:

- **Code generation:** generate a complete solution from the task description.

- **Full Repair:** given the buggy code, produce a corrected version.

- **Diff Repair:** produce a patch (diff) that modifies the buggy code to fix the error.

- **Prompt with test cases [our default]:** given the buggy code and the unit test error traces, produce a corrected version.

The performance results are included in Table 3. We find that test feedback improves repair pass rate as the traces help in locating some of the causes and types of bugs, while diff-patch based repair often yields lower performance due to the complexity of the format and of repairing code with more restricted output. For Diff-Repair, we implement a custom diff applier which handles fuzzy context matching to best handle the model outputs. In our main experiments we opted to use the testcases-based approach, motivated by its improved performance as well as its alignment with most real-world code repair settings.

*Table 4.* Fix rates (%) by dataset and fixer model.

| Outcome | Human | | | Human-Edited LM | | |
|---|---|---|---|---|---|---|
| | o4-mini | GPT-5.2 | Sonnet | o4-mini | GPT-5.2 | Sonnet |
| Fix Fail \| Code Pass | 16.4 | 16.9 | 8.2 | 27.9 | 27.1 | 41.0 |
| Fix Pass \| Code Fail | 59.1 | 54.4 | 71.2 | 42.4 | 36.8 | 43.9 |
| | LM Errors (Qwen-7B) | | | LM Errors (gpt-oss-20b) | | |
| | o4-mini | GPT-5.2 | Sonnet | o4-mini | GPT-5.2 | Sonnet |
| Fix Fail \| Code Pass | 42.6 | 39.0 | 41.0 | 41.0 | 28.8 | 39.3 |
| Fix Pass \| Code Fail | 33.3 | 30.9 | 31.8 | 40.9 | 32.4 | 30.3 |

**Debugging is a distinct skill from code generation.** Solving a task from scratch and repairing an existing solution succeed on different instances. In Table 4, we can observe that, for instance, Claude-Sonnet on BUGSOURCEBENCH-Human-Edited LM has a sizable fraction of cases where code generation passes but repair fails (0.410), and many cases where repair succeeds but code generation fails (0.439). Similar patterns hold across models and splits, indicating that repair requires capabilities, such as localizing faults, that are not captured by end-to-end code generation.

# B. Training Details

## B.1. ANCHORED SELF-PLAY (ASP) & Self-Play Hyperparameters

We train a Qwen2.5-Coder-7B-Instruct policy with GRPO under the following settings.

- **Optimization and regularization.** Learning rate $1 \times 10^{-6}$. PPO-style clipping with ratio $0.28$.
- **Batching.** Training batch size 64. Validation batch size 256. PPO minibatch size 32. Dynamic batch sizing enabled. Maximum PPO token budget of 30,000 tokens per GPU.
- **Sequence lengths.** Maximum prompt length 8192 tokens. Maximum response length 2048 tokens.
- **Rollouts.** Asynchronous rollouts. Sampling temperature 0.6 for training and validation. Top-$p$ 0.95 for validation. Samples per prompt: $n = 4$ for training and $n = 1$ for validation.
- **Systems settings.** Gradient checkpointing enabled.
- **Training schedule and logging.** 10 epochs total.
- **Parallelism and hardware.** One node with 8 GPUs. Tensor parallel size 1 and sequence parallel size 1.

**Self-play loop hyperparameters.** For each task, we sample $G = 4$ candidate bugs. For each bug, we sample $K = 4$ independent repair attempts and compute the solve rate $\rho$. We use band-shaped generator rewards with $\rho_\ell = 0.25$, $\rho_h = 0.75$, invalid-bug reward $-1.0$, and extreme-case penalty $\alpha = 0.2$. We include failing test output in the fixer context and normalize advantages separately for the generator and fixer roles.

**Anchoring hyperparameters.** For ANCHORED SELF-PLAY (ASP), we use a reference dataset from all BUG-SOURCEBENCH splits: Human, Human-Edited LM, LM Errors (Qwen-7B), LM Errors (gpt-oss-20b). We enable embedding-similarity shaping for the generator with weight $\lambda = 0.20$ and use embedding scores from `voyage-code-3` on diffs between the buggy program and reference program. We compute similarity using diff-based edit embeddings and a $k$-nearest-neighbor score with $k = 5$. We use margin scoring with temperature $5.0$ and an exponential-moving-average baseline with decay $\beta = 0.99$. In each fixer training batch, 20% of the samples are from the reference bugs, randomly sampled from the reference dataset.

### B.2. Fixer-Only Hyperparameters

We train a Qwen2.5-Coder-7B-Instruct *fixer* with GRPO while keeping the *generator* frozen and served externally. To approximately control for compute relative to Self-Play and ASP, we use $n = 16$ actor rollouts per training prompt (validation uses $n = 1$).

**Data and prompting hyperparameters.** To align supervision between Fixer-Only and ASP, we train on a mixture of BigCodeBench and the training splits of the reference bug-source datasets (i.e., the same reference/target data available to ASP). During repair, the fixer input includes the failed unit-test output when available. At validation time, we evaluate both repair (given a buggy program) and standard code generation.

- **Optimization and regularization.** Learning rate $1 \times 10^{-6}$. PPO-style clipping with ratio $0.28$.
- **Batching.** Training batch size $64$. Validation batch size $256$. PPO minibatch size $32$. Dynamic batch sizing enabled. Maximum PPO token budget of $24{,}000$ tokens per GPU.
- **Sequence lengths.** Maximum prompt length $8192$ tokens. Maximum response length $2048$ tokens.
- **Rollouts.** Asynchronous rollouts. Sampling temperature $0.6$ for training and validation. Top-$p$ $0.95$ for both the frozen generator and validation sampling. Samples per prompt: $n = 16$ for training and $n = 1$ for validation.
- **Frozen generator configuration.** Generator model: Qwen2.5-Coder-7B-Instruct. Generation temperature $0.6$ and top-$p$ $0.95$.
- **Training schedule and logging.** 10 epochs total.
- **Parallelism and hardware.** One node with 8 GPUs. Tensor parallel size 1 and sequence parallel size 1.

**Compute.** All runs use a single node with 8 H100 GPUs and take approximately 48 hours for 120 training steps.

## C. Prompts

We use two roles: a *bug generator* that injects subtle faults into a correct reference solution, and a *bug fixer* that repairs the buggy program using unit-test feedback when available. In the prompts below, `<PROBLEM>` and `` denote placeholders filled at runtime.

**Bug generator prompt**

```
1   You are a *bug generator* for Python solutions to competitive programming problems.
2
3   You will be given:
4
5   1. A problem description.
6   2. A *correct* reference implementation in Python.
7
8   Your task:
9
10  - Introduce **one or a few subtle bugs** into the code.
11  - The resulting code **must still be syntactically valid Python**.
```

```
12  - It should change the behavior so that **at least one unit test fails**.
13  - Do **not** drastically rewrite the code; keep the overall structure similar.
14  - Do **not** change the function signature, imports, or I/O format.
15  - Output **only** the full buggy Python code inside a single '''python''' block.
16
17  Problem:
18  <PROBLEM>
19
20  Correct reference implementation:
21  <CORRECT_CODE>
22
23  Now generate the buggy version of this code. Return the entire function with the
        buggy code inside a '''python''' block:
```

**Bug fixer prompt**

```
1   You are an expert Python debugging assistant.
2
3   You will be given:
4
5   1. A problem description.
6   2. A buggy Python implementation that may fail some hidden unit tests.
7   3. (Optional) Failed unit test output from running the buggy implementation.
8
9   Your task:
10
11  - Carefully read the code and identify the bug(s).
12  - Produce a fixed version of the code that makes all unit tests pass.
13  - Preserve the original function signature, imports, and I/O format.
14  - Keep the solution reasonably close to the given implementation.
15  - Output **only** the full corrected Python code inside a single '''python''' block.
16
17  Problem:
18  <PROBLEM>
19
20  Buggy implementation:
21  <BUGGY_CODE>
22
23  Failed unit test output (if available):
24  '''text
25  <FAILED_TEST_OUTPUT>
26  '''
27  Now fix the bugs in this code. Return the entire function with the fixed code inside
        a '''python''' block:
```

# D. Additional Evaluations

### D.1. Pass@k Results

We report pass@k with temperature sampling across all methods and bug sources. ASP outperforms all baselines at every sampling budget, and the gap widens at higher $k$.

### D.2. Qualitative Examples of Generated Bugs

Figure 8 presents an example of a bug generated by ASP.

*Table 5.* Pass@k results on BUGSOURCEBENCH (averaged across bug sources).

| Method | pass@1 | pass@5 | pass@10 |
|---|---|---|---|
| Base Model | 25.8 | 37.3 | 43.8 |
| Fixer-Only | 27.9 | 40.4 | 46.2 |
| Self-Play | 29.1 | 41.5 | 48.3 |
| **ASP** (Ours) | **36.1** | **50.7** | **57.2** |

---

**BugSourceBench TaskID 953**

Generates random data points to plot bar charts for each in a given list of plot names, then saves them in a specified directory. This function takes a list of plot names, for each generating 10 random data points in [0, 1) to create a bar chart, then saves the bar charts as .png files in the specified directory, creating the directory if it does not exist.

```
>> task_func(['First Plot', 'Second Plot'], './another_folder/')
['First_Plot.png', 'Second_Plot.png']
```
*Note that:* This function deduplicates `mystrings` while maintaining its original order. Random data points for bar charts are generated in the range [0, 1). Each bar chart contains 10 data points.

*The function should raise the exception for:* `FileNotFoundError`: If the provided directory path does not exist and cannot be created.
*The function should output with:* list: Names of the files where the plots are saved. Each file corresponds to a title from `mystrings`.

**BASE**

```python
import numpy as np
import matplotlib.pyplot as plt
import os

def task_func(mystrings, folder_path, seed=None):
    if seed is not None:
        np.random.seed(seed)
    saved_plots = []
    processed_names = set()

    if not os.path.exists(folder_path):
        os.mkdir(folder_path)
    for name in mystrings:
        if name in processed_names:
            continue

        data = np.random.rand(10)
        plt.bar(range(len(data)), data)
        plt.title(name)
        file_name = name.replace(" ", "_") + ".png"
        plt.savefig(os.path.join(folder_path, file_name))

        saved_plots.append(file_name)
        processed_names.add(name)
    return saved_plots
```

**ASP**

```python
import numpy as np
import matplotlib.pyplot as plt
import os

def task_func(mystrings, folder_path, seed=None):
    if seed is not None:
        np.random.seed(seed)

    saved_plots = []

    if not os.path.exists(folder_path):
        raise Exception("Directory could not be created.")

    for name in mystrings:
        data = np.random.rand(10)

        plt.bar(range(len(data)), data)
        plt.title(name)
        file_name = name.replace(" ", "_") + ".png"
        plt.savefig(os.path.join(folder_path, file_name))
        saved_plots.append(file_name)

    return ["error"]
```

*Figure 8.* BUGSOURCEBENCH example showing bugs generated by the model before and after anchored self-play training. The base model generates a very simple and artificial typo, and after training creates more logical errors, with missing de-duplication, improper error handling, and failing to follow instructions in creating a new directory.

## D.3. Similarity of Generated Bugs

**Mean similarity evolution.** For each generator checkpoint and method (Self-Play vs. ASP), we sample a large set of synthetic bugs on the tasks from the test set. We embed each generated bug (diff-embedding) and compute its similarity to a chosen target bug pool (e.g., BugSourceBench source split) as the mean top-(k) cosine similarity to the pool embeddings (excluding same-task matches when available). We then report the mean similarity across generated bugs as a function of training step, producing a "similarity evolution" curve that tracks how the generator's output distribution moves relative to each target pool over training.

**Target fix rate by generation-similarity quantile.** To test whether downstream gains are explained solely by generating more target-like bugs, we construct global similarity quantile bins using the generator's similarities to a target pool (shared bin boundaries across methods). For each bin, we evaluate fixer performance on the *test* BUGSOURCEBENCH *instances* whose task IDs correspond to that bin, reporting the target test solve rate (with confidence intervals) per quantile. This yields a "matched-similarity" diagnostic: it compares ASP vs. Self-Play at comparable levels of generator–target similarity, allowing us to assess whether ASP improves target robustness beyond simply shifting the generator toward higher-similarity bugs.

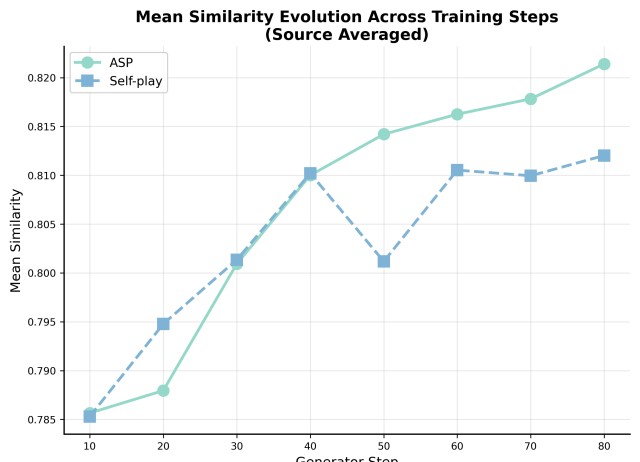

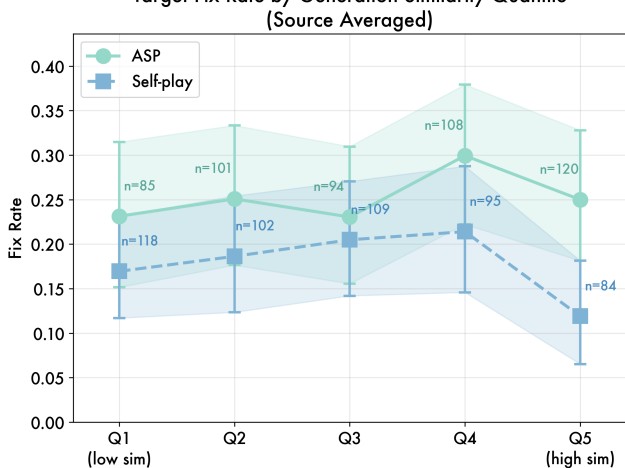

*(a)* **Similarity-guided shaping increases target-likeness of generated bugs.**

*(b)* **Performance on test tasks by generated similarity.**

*Figure 9.* We sample $n = 3$ bugs from the generator for the tasks in the held-out BugSourceBench test and plot the kNN embedding score of the generations to the target bugs (higher is more target-like). Similarity-guided shaping in ASP yields a consistent increase over training compared to standard self-play, indicating generator outputs move toward the target bug distribution under the shaping signal (left). Next, we bucket tasks into similarity quantiles based on their generated bugs, sample $k = 3$ fix attempts per test task, and report fix rate as a function of the task's corresponding similarity bucket with 95% confidence intervals (right).

### D.4. Evaluation by Semantic Bug Type

We categorize bugs by semantic type (independent of generating model) and report fix rates per category in Table 6. ASP improves consistently across categories, suggesting gains stem from stronger repair capability rather than adaptation to specific error patterns.

*Table 6.* Fix rate (%) by semantic bug type.

| Model | Logic | Wrong Val. | Edge Case | API Misuse | Other |
|---|---|---|---|---|---|
| Qwen-7B (base) | 14 | 37 | 5 | 20 | 33 |
| Fixer-Only | 12 | 37 | 5 | 14 | 28 |
| Self-Play | 16 | 37 | 9 | 18 | 44 |
| **ASP** (Ours) | **21** | 36 | **12** | **29** | 38 |

### D.5. Evaluation as a Test-Time Fixer

We evaluate whether ASP-trained fixers can improve larger code generation models at test time. We use DeepSeek-Coder-33B and Qwen2.5-Coder-32B as coders, generate initial solutions, then apply 7B fixers iteratively. Results are reported on corresponding BigCodeBench problems (from BugSourceBench test splits) with $k$ samples and $r$ repair rounds in Table 7. ASP fixers consistently outperform self-play and fixer-only fixers, and improve over self-repair with the large coder model itself.

## E. Additional Experiments

### E.1. Additional Baselines

We present results for several additional baseline experiments.

**Reinforcement Learning for Code Generation.** We show that code generation capabilities are distinct from code repair. In order to present an ablation baseline on the impact of training for code generation capabilities, we train the base model on the task of code generation given the task specs with binary rewards based on testcase execution; we follow aligned settings

*Table 7.* **Test-time fixer results.** Pass rate (%) using large coders with fixer models over $r$ repair rounds and $k$ samples. The *No repair* row is the coder alone ($r=0$); all other rows apply the listed fixer.

| | $r=1$ | | $r=2$ | |
|---|---|---|---|---|
| **Fixer** | $k=1$ | $k=2$ | $k=1$ | $k=2$ |
| Coder: DeepSeek-33B *No repair: k=1: 29.5* | | *k=2: 37.0* | | |
| Qwen-7B (base) | 35.4 | 40.9 | 36.2 | 47.2 |
| Fixer-Only | 34.6 | 41.7 | 35.4 | 41.7 |
| Self-Repair (Qwen-32B) | 34.6 | 42.5 | 37.8 | 49.6 |
| Self-Play | 35.4 | 42.5 | 37.8 | 46.5 |
| **ASP** (Ours) | **36.2** | **45.7** | **43.3** | **53.5** |
| Coder: Qwen-32B    *No repair: k=1: 30.3* | | *k=2: 37.0* | | |
| Qwen-7B (base) | 33.1 | 40.9 | 36.2 | 44.1 |
| Fixer-Only | 33.9 | 40.2 | 37.0 | 43.3 |
| Self-Repair (DeepSeek-33B) | 35.4 | 40.9 | **44.9** | **57.5** |
| Self-Play | 33.9 | 43.3 | 37.8 | 48.0 |
| **ASP** (Ours) | **38.6** | **44.1** | **44.9** | 54.3 |

and training data splits to our standard repair experiments.

**Self-Play with Constrained Bug Generation**    Our main experiments focus on open-ended bug generation for self-play. An alternative anchoring approach depends on constrained generation. We provide a constrained bug generation baseline that restricts the generator to AST mutations, which structurally modify the syntax tree representations of the code while maintaining syntactic integrity and producing bugs with compiling code. We identify and provide valid AST mutation sites in the original reference code implementation to the generator model and allow it to return valid AST mutations to apply to produce buggy code. We use bugs from the constrained generator and perform self-play as usual.

**Supervised Fine-Tuning for Fixer Training.**    We present performance for training the fixer with SFT. We train for the task of fixing buggy code from the train instances of BUGSOURCEBENCH across all 4 sources and use a masked cross-entropy loss on the corresponding correct reference solutions.

**Anchored Self-Play with Alternate Embeddings.**    We demonstrate the robustness of ASP to alternate embedding choices, presenting an experiment following the standard ASP setup with `CodeBERT` embeddings instead of `voyage-code-3` embeddings.

*Table 8.* **Fix rates for additional baselines.** For the BUGSOURCEBENCH splits, we report fix rates (in %), and for the codegen interface we report code generation pass rate.

| Method | Codegen | Human | Human-Ed. | LM (Qwen-7B) | LM (gpt-oss-20b) |
|---|---|---|---|---|---|
| Base Model | 21.7 | 58.3 | 30.8 | 7.1 | 7.0 |
| Codegen RL | **31.8** | 55.8 | 30.8 | 2.9 | 6.3 |
| AST-Constrained Self-Play | – | 59.8 | 26.8 | 3.9 | 3.1 |
| Fixer SFT | – | 46.5 | 18.1 | 10.2 | 11.0 |
| ASP (`CodeBERT`) | – | **63.3** | **33.9** | **23.6** | **24.6** |

Results in Table 8 show that while RL post-training on code generation improves code generation capabilities, performance on repair tasks stays stable or even degrades. The SFT-trained Fixer-Only model degrades performance on Human and Human-Edited LM splits, providing minimal improvements.

### E.2. Additional Ablations

We present several additional ablation experiments.

**Embedding and Similarity Ablations**    We ablate the kNN pooling parameter $k$ and the embedding model used for the similarity reward (note `CodeBERT` baseline in Appendix E.1) in Table 9. ASP is stable across moderate $k$ and is not tied to a single embedding model.

*Table 9.* kNN pooling $k$ (voyage-code-3) and embedding model ablations.

| $k$ | Avg Fix Rate (%) |
|---|---|
| 1 | 34.7 |
| 5 (default) | 36.1 |
| 10 | 35.8 |
| 20 | 35.0 |

| Embedding | Avg Fix Rate (%) |
|---|---|
| voyage-code-3 (default) | 36.1 |
| CodeBERT | 36.4 |

**Updating the Generator on Mixed Episodes** Reference mixing replaces a generated bug with a reference bug for fixer training. We do not update the generator on these mixed episodes because the generator's usual difficulty-band reward is defined for its own generated bug; assigning a generator update on a reference-mixed episode either makes the update off-policy or requires additional fixer rollouts, breaking compute matching. We nevertheless ran this ablation in Table 10. Updating the generator on mixed episodes reduces average fix rate from 36.1% to 33.6%, supporting our choice to keep generator updates on-policy while using reference bugs to train the fixer.

*Table 10.* Ablation that updates the generator on reference-mixed episodes. Fix rates are reported on BUGSOURCEBENCH splits.

| Method | Human | Human-Ed. | LM (Qwen-7B) | LM (gpt-oss) | Avg |
|---|---|---|---|---|---|
| Base Model | 58.3 | 30.8 | 7.1 | 7.0 | 25.8 |
| Fixer-Only | 58.3 | 30.7 | 10.2 | 12.6 | 27.9 |
| Self-Play | 63.7 | 31.5 | 8.7 | 12.6 | 29.1 |
| **ASP** (Ours) | **65.0** | **37.0** | **21.3** | **21.3** | **36.1** |
| ASP + gen on mixed | 64.1 | 35.8 | 16.1 | 18.3 | 33.6 |

**Decoupled Generator and Fixer** We test whether drift and ASP's gains are artifacts of weight sharing by training separate models for generation and fixing. Results in Table 11 show drift is *worse* with decoupled weights (Human: 28.5 vs. 31.7 with shared), while ASP's gains are robust to decoupling (34.8 vs. 35.5). This confirms drift is a property of the reward signal, not of weight sharing.

*Table 11.* Shared vs. decoupled generator/fixer weights.

| Method | Weights | Avg | Human | Human-Edited LM | LM (Qwen-7B) | LM (gpt-oss-20b) |
|---|---|---|---|---|---|---|
| Self-Play | shared | 20.9 | 31.7 | 24.4 | 12.6 | 15.0 |
| Self-Play | decoupled | 19.6 | 28.5 | 22.5 | 12.0 | 15.5 |
| ASP | shared | 35.5 | 64.2 | 37.0 | 18.9 | 22.0 |
| ASP | decoupled | 34.8 | 61.5 | 36.0 | 18.0 | 23.5 |

**Alternate Base Model** We replicate our experiments with DeepSeek-Coder-6.7B-Instruct as the base model in Table 12. Self-play again shows limited gains and slight regression on human bugs, while ASP improves all sources (+8.9 pp over self-play on average).

**External Benchmarks and Task Distributions.** Beyond BUGSOURCEBENCH, we also evaluate performance on an external benchmark, DebugBench (Tian et al., 2024), which tests debugging capabilities with buggy solutions to problems derived from LeetCode in multiple programming languages. We evaluate on the 1340 buggy codes in Python using our standard fixer prompt format and settings in our BUGSOURCEBENCH evaluations. We demonstrate the robustness of our approach on improving performance across distributions through these external benchmark results in Table 13. Even on target distributions distinct from the reference set, ASP outperforms Self-play, suggesting better generalization to OOD bug distributions.

In addition to the transfer results in Table 13 (BigCodeBench-trained models evaluated on DebugBench), we also demonstrate the persistence of drift in other task distributions by training Self-Play and ASP directly on a split of DebugBench tasks.

*Table 12.* Results with DeepSeek-Coder-6.7B-Instruct as base model.

| Method | Human | Human-Edited LM | LM (Qwen-7B) | LM (gpt-oss-20b) | Avg |
|---|---|---|---|---|---|
| Base Model | 51.6 | 22.8 | 3.7 | 2.6 | 20.2 |
| Fixer-Only | 51.6 | 22.7 | 6.8 | 8.2 | 22.3 |
| Self-Play | 50.8 | 22.0 | 5.3 | 8.2 | 21.6 |
| **ASP** (Ours) | **58.3** | **29.0** | **17.9** | **16.9** | **30.5** |

*Table 13.* **Fix rates (%) on DebugBench (Tian et al., 2024).**

| Method | DebugBench |
|---|---|
| Base Model | 55.0 |
| Codegen RL | 54.3 |
| AST-Constrained Self-Play | 56.6 |
| Fixer SFT | 56.0 |
| Fixer-Only | 54.6 |
| Self-Play | 58.7 |
| ASP (CodeBERT) | 59.0 |
| **ASP** (Ours) | **60.3** |

In Table 14, Self-play exhibits drift on DebugBench held-out bugs while ASP stably improves, confirming drift is a fundamental property of unit-test-only self-play, not specific to BigCodeBench.

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

*Table 14.* Held-out DebugBench fix rate (%) when trained on DebugBench tasks.

| Method | Fix Rate |
|---|---|
| Qwen-7B (base) | 58.0 |
| Self-Play | 58.7 |
| **ASP** (Ours) | **66.3** |

Dou, S., Jia, H., Wu, S., Zheng, H., Wu, M., Tao, Y., Zhang, M., Chai, M., Fan, J., Xi, Z., Zheng, R., Wu, Y., Wen, M., Gui, T., Zhang, Q., Qiu, X., and Huang, X. What's wrong with your code generated by large language models? an extensive study, 2025. URL https://arxiv.org/abs/2407.06153.

Forrest, S., Nguyen, T., Weimer, W., and Le Goues, C. A genetic programming approach to automated software repair. In *Proceedings of the 11th Annual Conference on Genetic and Evolutionary Computation (GECCO)*, 2009.

He, J., Beurer-Kellner, L., and Vechev, M. On distribution shift in learning-based bug detectors. In *International conference on machine learning*, pp. 8559–8580. PMLR, 2022.

Hendrycks, D., Basart, S., Kadavath, S., Mazeika, M., Arora, A., Guo, E., Burns, C., Puranik, S., He, H., Song, D., et al. Measuring coding challenge competence with apps. In *Advances in Neural Information Processing Systems*, 2021.

Huang, C., Yu, W., Wang, X., Zhang, H., Li, Z., Li, R., Huang, J., Mi, H., and Yu, D. R-zero: Self-evolving reasoning llm from zero data. *arXiv preprint arXiv:2508.05004*, 2025.

Hui, B., Yang, J., Cui, Z., Yang, J., Liu, D., Zhang, L., Liu, T., Zhang, J., Yu, B., Lu, K., et al. Qwen2. 5-coder technical report. *arXiv preprint arXiv:2409.12186*, 2024.

Jain, N., Han, K., Gu, A., Li, W.-D., Yan, F., Zhang, T., Wang, S., Solar-Lezama, A., Sen, K., and Stoica, I. Livecodebench: Holistic and contamination free evaluation of large language models for code. 2024. URL https://arxiv.org/abs/2403.07974.

Jimenez, C. E., Yang, J., Wettig, A., Yao, S., Pei, K., Press, O., and Narasimhan, K. Swe-bench: Can language models resolve real-world github issues? *arXiv preprint arXiv:2310.06770*, 2023.

Just, R., Jalali, D., and Ernst, M. D. Defects4j: A database of existing faults to enable controlled testing studies for java programs. In *Proceedings of the 2014 international symposium on software testing and analysis*, pp. 437–440, 2014.

Kuba, J. G., Gu, M., Ma, Q., Tian, Y., Mohan, V., and Chen, J. Language self-play for data-free training. *arXiv preprint arXiv:2509.07414*, 2025.

Le Goues, C., Holtschulte, N., Smith, E. K., Brun, Y., Devanbu, P., Forrest, S., and Weimer, W. The manybugs and introclass benchmarks for automated repair of c programs. *IEEE Transactions on Software Engineering*, 41(12):1236–1256, 2015.

Li, H., Tang, Y., Wang, S., and Guo, W. Patchpilot: A stable and cost-efficient agentic patching framework. *arXiv preprint arXiv:2502.02747*, 2025. URL https://arxiv.org/abs/2502.02747.

Lin, D., Koppel, J., Chen, A., and Solar-Lezama, A. Quixbugs: A multi-lingual program repair benchmark set based on the quixey challenge. In *Proceedings Companion of the 2017 ACM SIGPLAN international conference on systems, programming, languages, and applications: software for humanity*, pp. 55–56, 2017.

Lin, Z., Shen, S., Shang, J., Weston, J., and Nie, Y. Learning to solve and verify: A self-play framework for code and test generation. *arXiv preprint arXiv:2502.14948*, 2025.

Liu, B., Jin, C., Kim, S., Yuan, W., Zhao, W., Kulikov, I., Li, X., Sukhbaatar, S., Lanchantin, J., and Weston, J. Spice: Self-play in corpus environments improves reasoning. *arXiv preprint arXiv:2510.24684*, 2025.

Liu, J., Xia, C. S., Wang, Y., and Zhang, L. Is your code generated by ChatGPT really correct? rigorous evaluation of large language models for code generation. In *Advances in Neural Information Processing Systems*, volume 36, 2023. URL https://arxiv.org/abs/2305.01210. arXiv:2305.01210.

Long, F. and Rinard, M. Automatic patch generation by learning correct code. In *Proceedings of the 43rd Annual ACM SIGPLAN-SIGACT Symposium on Principles of Programming Languages (POPL)*, 2016.

Madeiral, F., Urli, S., Maia, M., and Monperrus, M. Bears: An extensible Java bug benchmark for automatic program repair studies. In *Proceedings of the 26th IEEE International Conference on Software Analysis, Evolution and Reengineering (SANER)*, pp. 468–478, 2019. doi: 10.1109/SANER.2019.8667991. URL https://arxiv.org/abs/1901.06024.

Oliva, G. A., Rajbahadur, G. K., Bhatia, A., Zhang, H., Chen, Y., Chen, Z., Leung, A., Lin, D., Chen, B., and Hassan, A. E. Spice: An automated swe-bench labeling pipeline for issue clarity, test coverage, and effort estimation. *arXiv preprint arXiv:2507.09108*, 2025.

Parker-Holder, J., Jiang, M., Dennis, M., Samvelyan, M., Foerster, J., Grefenstette, E., and Rocktäschel, T. Evolving curricula with regret-based environment design. 2023. URL https://arxiv.org/abs/2203.01302.

Pham, M. V., Phan, H. N., Phan, H. N., Chi, C. L., Nguyen, T. N., and Bui, N. D. Swe-synth: Synthesizing verifiable bug-fix data to enable large language models in resolving real-world bugs. *arXiv preprint arXiv:2504.14757*, 2025.

Poesia, G., Broman, D., Haber, N., and Goodman, N. D. Learning formal mathematics from intrinsic motivation. 2024. URL https://arxiv.org/abs/2407.00695.

Pourcel, J., Colas, C., Molinaro, G., Oudeyer, P.-Y., and Teodorescu, L. Aces: Generating a diversity of challenging programming puzzles with autotelic generative models. *Advances in Neural Information Processing Systems*, 37: 67627–67662, 2024.

Ribeiro, M. T. and Lundberg, S. Adaptive testing and debugging of NLP models. In *Proceedings of the 60th Annual Meeting of the Association for Computational Linguistics (Volume 1: Long Papers)*, pp. 3253–3267, Dublin, Ireland, May 2022. Association for Computational Linguistics. doi: 10.18653/v1/2022.acl-long.230. URL https://aclanthology.org/2022.acl-long.230/.

Silver, D., Schrittwieser, J., Simonyan, K., Antonoglou, I., Huang, A., Guez, A., Hubert, T., Baker, L., Lai, M., Bolton, A., et al. Mastering the game of go without human knowledge. *nature*, 550(7676):354–359, 2017.

Sonwane, A., White, I., Lee, H., Pereira, M., Caccia, L., Kim, M., Shi, Z., Singh, C., Sordoni, A., Côté, M.-A., and Yuan, X. Bugpilot: Complex bug generation for efficient learning of swe skills, 2025. URL https://arxiv.org/abs/2510.19898.

Tang, Y., Li, H., Zhu, K., Yang, M., Ding, Y., and Guo, W. Co-PatcheR: Collaborative software patching with component(s)-specific small reasoning models. *arXiv preprint arXiv:2505.18955*, 2025. URL https://arxiv.org/abs/2505.18955.

Teodorescu, L., Colas, C., Bowers, M., Carta, T., and Oudeyer, P.-Y. Codeplay: Autotelic learning through collaborative self-play in programming environments. In *IMOL 2023: Intrinsically Motivated Open-ended Learning Workshop at NeurIPS 2023*, 2023. URL https://openreview.net/forum?id=GgzfIBxa18.

Tian, R., Ye, Y., Qin, Y., Cong, X., Lin, Y., Pan, Y., Wu, Y., Hui, H., Liu, W., Liu, Z., and Sun, M. Debugbench: Evaluating debugging capability of large language models, 2024. URL https://arxiv.org/abs/2401.04621.

Wang, X., Li, B., Song, Y., Xu, F. F., Tang, X., Zhuge, M., Pan, J., Song, Y., Li, B., Singh, J., et al. OpenHands: An open platform for ai software developers as generalist agents. In *International Conference on Learning Representations*, 2025a. URL https://arxiv.org/abs/2407.16741.

Wang, Y., Yang, L., Tian, Y., Shen, K., and Wang, M. Cure: Co-evolving coders and unit testers via reinforcement learning. In *The Thirty-ninth Annual Conference on Neural Information Processing Systems*, 2025b.

Wei, Y., Sun, Z., McMilin, E., Gehring, J., Zhang, D., Synnaeve, G., Fried, D., Zhang, L., and Wang, S. Toward training superintelligent software agents through self-play swe-rl. *arXiv preprint arXiv:2512.18552*, 2025.

Widyasari, R., Sim, S. Q., Lok, C., Qi, H., Phan, J., Tay, Q., Tan, C., Wee, F., Tan, J. E., Yieh, Y., et al. Bugsinpy: a database of existing bugs in python programs to enable controlled testing and debugging studies. In *Proceedings of the 28th ACM joint meeting on european software engineering conference and symposium on the foundations of software engineering*, pp. 1556–1560, 2020.

Wilf, A., Aggarwal, P., Parno, B., Fried, D., Morency, L.-P., Liang, P. P., and Welleck, S. Propose, solve, verify: Self-play through formal verification. *arXiv preprint arXiv:2512.18160*, 2025.

Xia, C. S., Deng, Y., Dunn, S., and Zhang, L. Agentless: Demystifying llm-based software engineering agents. *arXiv preprint arXiv:2407.01489*, 2024. URL https://arxiv.org/abs/2407.01489.

Xu, F. F., Alon, U., Neubig, G., and Hellendoorn, V. J. A systematic evaluation of large language models of code. In *Proceedings of the 6th ACM SIGPLAN international symposium on machine programming*, pp. 1–10, 2022.

Yang, J., Jimenez, C. E., Wettig, A., Lieret, K., Yao, S., Narasimhan, K., and Press, O. SWE-agent: Agent-computer interfaces enable automated software engineering. In *Advances in Neural Information Processing Systems*, 2024. URL https://arxiv.org/abs/2405.15793.

Yang, J., Leret, K., Jimenez, C. E., Wettig, A., Khandpur, K., Zhang, Y., Hui, B., Press, O., Schmidt, L., and Yang, D. Swe-smith: Scaling data for software engineering agents. *arXiv preprint arXiv:2504.21798*, 2025a.

Yang, W., Wang, H., Liu, Z., Li, X., Yan, Y., Wang, S., Gu, Y., Yu, M., Liu, Z., and Yu, G. Coast: Enhancing the code debugging ability of llms through communicative agent based data synthesis, 2025b. URL https://arxiv.org/abs/2408.05006.

Yasunaga, M. and Liang, P. Break-it-fix-it: Unsupervised learning for program repair. In *International Conference on Machine Learning (ICML)*, 2021.

Ye, H., Martinez, M., Luo, X., Zhang, T., and Monperrus, M. Selfapr: Self-supervised program repair with test execution diagnostics. In *Proceedings of the 37th IEEE/ACM International Conference on Automated Software Engineering*, ASE '22, New York, NY, USA, 2023. Association for Computing Machinery. ISBN 9781450394758. doi: 10.1145/3551349.3556926. URL https://doi.org/10.1145/3551349.3556926.

Yu, W., Liang, Z., Huang, C., Panaganti, K., Fang, T., Mi, H., and Yu, D. Guided self-evolving llms with minimal human supervision. *arXiv preprint arXiv:2512.02472*, 2025.

Zhao, A., Wu, Y., Yue, Y., Wu, T., Xu, Q., Lin, M., Wang, S., Wu, Q., Zheng, Z., and Huang, G. Absolute zero: Reinforced self-play reasoning with zero data. *arXiv preprint arXiv:2505.03335*, 2025.

Zhuo, T. Y., Vu, M. C., Chim, J., Hu, H., Yu, W., Widyasari, R., Yusuf, I. N. B., Zhan, H., He, J., Paul, I., et al. Bigcodebench: Benchmarking code generation with diverse function calls and complex instructions. *arXiv preprint arXiv:2406.15877*, 2024.

Zirak, A. and Hemmati, H. Improving automated program repair with domain adaptation. *ACM Trans. Softw. Eng. Methodol.*, 33(3), March 2024. ISSN 1049-331X. doi: 10.1145/3631972. URL https://doi.org/10.1145/3631972.

