# OpenReview forum: "Anchoring Self-Play for Code Repair"
_ICML.cc/2026/Conference — ICML 2026 regular_

### Official Review · Reviewer_Pykf · 2026-03-11

**Soundness:** 1
**Presentation:** 1
**Significance:** 2
**Originality:** 2
**Overall Recommendation:** 2
**Confidence:** 4

**Summary:**

This paper studies whether code repair supervision can be scaled using synthetic bug generation via self-play. The authors propose a generator–fixer self-play framework where a language model alternates between generating buggy programs and repairing them using reinforcement learning. However, the authors observe that vanilla self-play suffers from distribution drift, producing synthetic bugs that differ from realistic human or model-generated bugs. To address this, the paper proposes Anchored Self-Play (ASP), which stabilizes training using two mechanisms: first, similarity-guided generator reward using code embeddings to encourage realistic bug patterns, and second, reference bug mixing during fixer training to maintain exposure to realistic bug distributions.

To evaluate robustness across bug sources, the authors introduce BUGSOURCEBENCH, a benchmark containing bugs from four sources: human-written, human-edited LM code, and two LM-generated bug distributions. Experiments show that ASP improves average repair rates across sources relative to vanilla self-play and fixer-only baselines.

**Compliance With Llm Reviewing Policy:**

Affirmed.

**Key Questions For Authors:**

1. How sensitive is ASP to the choice of embedding model used in the similarity reward?
2. How sensitive is ASP to the choice of the base model?
3. Would a learned discriminator or critic outperform the static embedding similarity heuristic?
4. Does ASP still improve results when scaling to larger models (e.g., 30B+ code models) or closed-source LLMs?
5. How does the method perform on repository-level repair tasks like SWE-bench?

**Strengths And Weaknesses:**

**Strength**

1. The underlying problem of automated code repair is important and of significant practical value.
2. The paper presents a clear and well-articulated problem formulation.


**Weakness**

1. Although the paper addresses a valuable problem, it lacks comparison with a right baseline. A baseline should include a top-tier close source LLM API and/or top-tier open LLM such as (DeepSeek, Kimi, etc). It is not clear why the author chose Qwen2.5-Coder-7B-Instruct.

2. The discussions around two key claims of the paper, namely reference bug mixing and code-embedding similarity reward, are somewhat sparse. More formalism is recommended. Also, the reasoning behind the choice of voyage-code-3 is expressed sufficiently.

3. BUGSOURCEBENCH focuses on short, unit-testable repair tasks and therefore differs substantially from repository-level benchmarks such as SWE-bench. While the benchmark enables controlled analysis of bug-source distribution shifts, it does not capture the complexity of real-world debugging scenarios that require reasoning over large codebases, modifying multiple files, and interacting with build systems. Consequently, it remains unclear whether the proposed approach would generalize to repository-scale repair settings. This limitation is particularly important because a large portion of recent code LLM research on issue resolution and bug fixing evaluates systems at the repository level.

4. As shown in Figure 3, the improvement in average fix rate between the base model and ASP is relatively modest (10.3 percentage points). Moreover, the improvement on the most practically relevant split—human-authored bugs—is even smaller (6.7 percentage points). Given the additional complexity introduced by Anchored Self-Play, these gains appear limited and raise questions about whether the proposed method provides a meaningful advantage over the base model.

5. It is unusual to define bug categories based on the underlying sampling models (e.g., Qwen-7B and GPT-OSS-20B) rather than the semantic characteristics of the bugs themselves. Categorizing bugs by their generating LLM conflates the source distribution with the bug type, which may introduce bias in the analysis and make it difficult to interpret whether improvements stem from better repair capability or simply better adaptation to model-specific error patterns. A more standard approach would categorize bugs according to their semantic properties (e.g., logic errors, API misuse, or edge-case handling), independent of the model that produced them.

6. The presentation of the paper would benefit from careful proofreading, as there appear to be several typographical and formatting issues throughout the manuscript.

---

> ### Author Rebuttal · Authors · 2026-03-31
>
> Thank you for your review! We address your concerns below. We respectfully note several points where existing results may have been overlooked.
>
> > (W1) Lacks comparison with a top-tier close source LLM and/or top-tier open LLM. Unclear why Qwen2.5-Coder-7B-Instruct.
>
> Table 3 (Appendix A.4) reports GPT-5.2, o4-mini, and Claude-Sonnet across all splits. Even frontier models exhibit bug-source sensitivity (e.g., Claude-Sonnet: 81.1% on Human vs. 44.9% on Qwen-7B Errors).
>
> ASP is a training method requiring RL over weights; the relevant comparison is between training regimes applied to the same model, not with closed source LLMs.
>
> We chose Qwen-7B as a strong open model tractable for academic RL, consistent with concurrent works ([Wang et al., NeurIPS 2025](https://arxiv.org/abs/2506.03136); [Zhao et al., NeurIPS 2025](https://arxiv.org/abs/2505.03335)).
>
> > (Q1/W2) How sensitive is ASP to the choice of embedding model used in the similarity reward?
>
> We ran ASP with CodeBERT and observed similar performance.
> |Embedding|Avg Fix Rate (%)|
> |--|--:|
> |voyage-code-3 (default)|36.1|
> |CodeBERT|36.4|
>
> > (W3/Q5) BUGSOURCEBENCH is function-level only. How does ASP perform on SWE-bench?
>
> We agree that repo-level evaluation is an important direction. We chose function-level because (1) repo-level self-play requires infrastructure beyond academic compute budgets ([SSR](https://arxiv.org/abs/2512.18552) uses 512 GPUs for 1 training run), and (2) drift is best studied isolated from agentic scaffolding confounds. This is consistent with recent code self-play works ([Lin et al., 2025](https://arxiv.org/abs/2502.14948); [Wang et al., NeurIPS 2025](https://arxiv.org/abs/2506.03136); [Zhao et al., NeurIPS 2025](https://arxiv.org/abs/2505.03335)). While SSR constrains injection to structured edits (code deletion, git reversion), we address the harder fully open-ended setting.
>
> > (W4) The improvement in average fix rate is relatively modest.
>
> We believe the most relevant comparison is ASP vs. self-play, not ASP vs. base. Standard self-play *regresses* on human bugs while improving on LM bugs. ASP is the only method that improves every source: $+7.2pp$ over self-play overall.
>
> ASP also improves across semantic bug categories (see W5 response) and compounds with test-time scaling, reaching $53.5%$ vs $46.5%$ for self-play with DeepSeek-33B coder (see Q4 response).
>
>
> > (W5) Unusual to define bug categories by model rather than semantics.
>
> Because we train on synthetically generated bugs, the main risk is overfitting to the generator's source distribution, which source-based evaluation directly measures. Our sources reflect LM-assisted programming, where bugs come from human code, LM code, and human edits to LM generated code. Semantic categories cut across sources; Figure 7a already provides this analysis.
>
> That said, we agree that per-category analysis helps interpret ASP's gains:
>
> |Model|LOGIC|WRONG_VAL|EDGE_CASE|API_MIS|
> |--|--:|--:|--:|--:|
> |Self-Play|16%|37%|9%|18%|
> |**ASP (Ours)**|**21%**|**36%**|**12%**|**29%**|
>
> ASP improves consistently across categories, suggesting gains stem from stronger repair rather than adaptation to specific error patterns.
>
>
> > (W6) Proofreading.
>
> Corrected in revision. We welcome specific suggestions for improvement.
>
> > (Q2) How sensitive is ASP to the choice of the base model?
>
> We ran with DeepSeek-Coder-6.7B-Instruct. We observe that self-play still shows limited gains, and ASP improves all sources ($+8.9pp$ over self-play).
>
> |Method|Human|Human-Ed.|Qwen-7B|gpt-oss-20b|Avg|
> |--|--:|--:|--:|--:|--:|
> |Base|51.6|22.8|3.7|2.6|20.2|
> |Fixer-Only|51.6|22.7|6.8|8.2|22.3|
> |Self-Play|50.8|22.0|5.3|8.2|21.6|
> |**ASP (Ours)**|58.3|29.0|17.9|16.9|30.5|
>
>
> > (Q3) Learned discriminator vs static embedding?
>
> A co-trained discriminator introduces a 3-way optimization prone to adversarial instabilities. Our static approach uses a frozen embedding as a fixed reference, analogous to [FID (Heusel et al., 2017)](https://arxiv.org/abs/1706.08500). We view learned critics as a promising extension (Section 9).
>
> > (Q4) Scaling to 30B+ or closed-source?
>
> Training at 30B+ is beyond our academic compute budget (each 7B run: 8 H100s, 48 hrs). We believe drift would persist at scale: it is fundamentally a Goodhart's law problem where the proxy reward does not capture realism, and more capable generators can exploit this gap ([Manheim & Garrabrant, 2019](https://arxiv.org/abs/1803.04585); [Gao et al., 2023](https://arxiv.org/abs/2210.10760)). Table 3 shows that even frontier models exhibit bug-source sensitivity. Hence we are optimistic that our method will also help larger models.
>
> That said, ASP-trained 7B models can improve larger models at test time. Using 30B+ coder models for code generation and our 7B ASP fixers for iterative repair (k=2 attempts, r=2 rounds):
>
> |Coder|Fixer|k=2, r=2|
> |--|--|--:|
> |DeepSeek-33B|Self-Play|46.5%|
> |DeepSeek-33B|**ASP**|**53.5%**|
> |Qwen-32B|Self-Play|48.0%|
> |Qwen-32B|**ASP**|**54.3%**|

---

> > ### Author Rebuttal · Reviewer_Pykf · 2026-04-02
> >
> > Thanks for your comments. However, I will keep my score. Thanks

---

> > > ### Author Response · Authors · 2026-04-03
> > >
> > > Thank you for your response and for reviewing our rebuttal. We appreciate your time and feedback.

---

### Official Review · Reviewer_AZqF · 2026-03-13

**Soundness:** 3
**Presentation:** 3
**Significance:** 2
**Originality:** 3
**Overall Recommendation:** 4
**Confidence:** 4

**Summary:**

This paper studies how to scale supervision for code repair via self-play, where a single model alternates between generating bugs and fixing them, using unit tests as the sole verification signal. The authors identify a key failure mode: vanilla self-play suffers from distribution drift, where the generator produces increasingly unrealistic bugs that break tests but diverge from bugs encountered in practice, causing the fixer to degrade on real-world bugs. To address this, they propose Anchored Self-Play (ASP), which anchors the self-play loop using a small reference set through two mechanisms: (i) an embedding-similarity reward that guides the generator toward reference-like bugs, and (ii) reference mixing that injects real bugs into fixer training. The paper also introduces BugSourceBench, a controlled benchmark with four bug sources (human-written, human-edited LM code, Qwen-7B errors, and gpt-oss-20b errors) sharing the same underlying tasks.

**Compliance With Llm Reviewing Policy:**

Affirmed.

**Key Questions For Authors:**

1. Have the authors considered evaluating on SWE-bench? Function-level repair and repo-level repair differ substantially in complexity, and it is unclear whether ASP's gains would transfer. Especially current LLM coding domain has already moved to agentic for a while.
2. Why does reference mixing not update the generator? Section 3.5.1 states that on mixed episodes, the generator is not updated "to preserve on-policy generation dynamics," but this explanation is brief and not backed by experiments. Have the authors tried updating the generator on mixed episodes as well — for example, providing the generator with a signal about what reference bugs look like? What specifically goes wrong if on-policy dynamics are not preserved? An ablation or at least a more detailed justification would be helpful.

3. It might be good to cite some code agent works:
[1] Swe-agent: Agent-computer interfaces enable automated software engineering.
[2] Agentless: Demystifying llm-based software engineering agents
[3] Patchpilot: A cost-efficient software engineering agent with early attempts on formal verification
[4] Co-patcher: Collaborative software patching with component (s)-specific small reasoning models
[5] Openhands: An open platform for ai software developers as generalist agents

**Limitations:**

1. The benchmark is limited to Python function-level tasks. BugSourceBench is built entirely on BigCodeBench, which consists of short, single-function Python problems with unit tests. This is far from real-world software engineering, where bugs span multiple files, involve complex dependencies, and require understanding repository context. The paper's title says "code repair" broadly, but the experimental coverage is narrow. The absence of experiments on repository-level benchmarks such as SWE-bench or Multi-SWE-bench is a notable gap (see Questions).
2. Evaluation uses only single greedy decoding. In practice, repair models are typically evaluated with multiple samples (pass@k). Reporting only greedy single-attempt results may not capture how different methods compare when given multiple chances, which is the more practical evaluation setting.

**Strengths And Weaknesses:**

1. The design of anchored self-play reflects a good understanding of both coding tasks and self-play training dynamics. The choice to anchor via edit-level embeddings (rather than, say, token-level similarity or syntactic constraints) shows an understanding of how code bugs manifest in practice — as semantic changes that are better captured in diff space.
2. BugSourceBench is thoughtfully designed. Sharing the same tasks across all bug sources while only varying the buggy program is a clean experimental design that removes task difficulty as a confound. The bug-type analysis and embedding-based clustering convincingly show that different sources produce systematically different bug patterns, validating the need for multi-source evaluation.
3. Ablations are thorough and internally consistent. The component ablation, reference pool composition, and reference set scaling together paint a coherent picture: both ASP components address complementary failure modes, and the reference set composition steers which bug patterns the model learns. The finding that code generation training does not improve (and may hurt) repair performance is also a useful insight.

---

> ### Author Rebuttal · Authors · 2026-03-31
>
> Thank you for your review and constructive feedback! We address your concerns below, and have revised our paper in response to your comments.
>
> > (Q1/L1) The benchmark is limited to function-level tasks
>
> We chose function-level tasks because (1) repo-level self-play requires significantly larger models and more training compute than available in academic settings (e.g., SWE-bench self-play [SSR (Wei et al., 2025)](https://arxiv.org/abs/2512.18552) uses 512 GPUs with a 131K context window), and (2) to isolate drift from confounds of agentic scaffolding.
>
> This scope is consistent with related code self-play works that operate at function level ([Lin et al., 2025](https://arxiv.org/abs/2502.14948); [Wang et al., NeurIPS 2025](https://arxiv.org/abs/2506.03136); [Wilf et al., 2025](https://arxiv.org/abs/2512.18160); [Zhao et al., NeurIPS 2025](https://arxiv.org/abs/2505.03335)).
>
> > (Q1/L1) Have the authors considered evaluating on SWE-bench?
>
> We agree that scaling to repo-level benchmarks and larger models is an important direction and a limitation of our work. SWE-bench evaluates the joint capability of localization, navigation, tool use, and repair, and extending ASP to this setting would strengthen the paper. However, repo-level self-play training requires substantial infrastructure — concurrent work on self-play for SWE-bench [SSR (Wei et al., 2025)](https://arxiv.org/abs/2512.18552) uses 512 GPUs for a single training run — which is beyond our academic compute budget. We chose to focus on cleanly diagnosing and addressing distribution drift at function level, where controlled experimentation is tractable.
>
> That said, SSR's approach constrains bug injection to structured edit types (code deletion, git history reversion) to prevent unrealistic bugs — implicitly acknowledging the drift problem we study. We address the fully open-ended setting where the generator can apply arbitrary text edits with unit tests as the sole constraint, making explicit anchoring necessary. ASP's components are granularity-agnostic and would complement repo-level approaches when environmental and edit-type constraints alone are insufficient.
>
> > (Q2) Why does reference mixing not update the generator? Section 3.5.1 states that on mixed episodes, the generator is not updated "to preserve on-policy generation dynamics"... Have the authors tried updating the generator on mixed episodes as well — for example, providing the generator with a signal about what reference bugs look like? An ablation or at least a more detailed justification would be helpful.
>
> We may be misunderstanding the question, so we address two possible interpretations:
>
> **Supervised imitation on reference bugs:** One interpretation is whether the generator should be trained to directly imitate the reference bugs. We do not do this because the reference bugs are written by humans or other LMs, rather than sampled from the generator’s current policy. Directly training on them would therefore be off-policy and could introduce compounding distribution mismatch ([Ross et al., 2011](https://arxiv.org/abs/1011.0686)). It would also increase the risk of overfitting, since the reference set is relatively small (~900 bugs). Instead, we use the similarity reward (Sec. 3.5.2), which provides an on-policy signal encouraging the generator toward the reference distribution in embedding space without requiring direct imitation.
>
> **RL updates on mixed episodes:** Another interpretation is whether the generator should still receive an RL update during mixed episodes. In these episodes, however, the fixer’s K rollouts are allocated to b_ref, so the generator’s usual difficulty-band reward for its own sample b_gen is not available. Recovering that reward would require running additional fixer rollouts on b_gen​, which would break compute matching with the self-play and fixer-only baselines.
>
> We nonetheless ran this ablation, which reduced average fix rate from 36.1 to 33.6:
>
> | Method | Human | Hum-Ed. | Qwen-7B | gpt-oss | Avg |
> |--------------------|------:|--------:|--------:|--------:|-----:|
> | Base | 58.3 | 30.8 | 7.1 | 7.0 | 25.8 |
> | Fixer-Only | 58.3 | 30.7 | 10.2 | 12.6 | 27.9 |
> | Self-Play | 63.7 | 31.5 | 8.7 | 12.6 | 29.1 |
> | ASP (Ours) | 65.0 | 37.0 | 21.3 | 21.3 | 36.1 |
> | ASP + gen on mixed | 64.1 | 35.8 | 16.1 | 18.3 | 33.6 |
>
> > (Q3) It might be good to cite some code agent works: [1] Swe-agent [2] Agentless [3] Patchpilot [4] Co-patcher [5] Openhands
>
> Thank you for the suggestion. We’ve added these citations.
>
> > (L2) Reporting pass@k
>
> Thank you for the suggestion. We’ve added pass@k results. ASP outperforms all baselines at higher sampling budgets.
>
> | Method     | pass@1 | pass@5 | pass@10 |
> |------------|-------:|-------:|--------:|
> | Base       | 25.8   | 37.3   | 43.8    |
> | Fixer-Only | 27.9   | 40.4   | 46.2    |
> | Self-Play  | 29.1   | 41.5   | 48.3    |
> | **ASP (Ours)** | **36.1**   | **50.7**   | **57.2**    |

---

> > ### Author Rebuttal · Reviewer_AZqF · 2026-04-03
> >
> > Thanks for the detailed response, which addresses a lot of my questions. The topic is good, and the method is creative; also, the experiment in the paper is detailed. Given limited resources, the decision to focus this work on function-level problems and evaluations based on open-source models is understandable; however, this also entails that its significance within the rapidly evolving field of LLMs will be somewhat limited. Overall, I will keep my positive attitude with the current score(week accept) for this paper.

---

> > > ### Author Response · Authors · 2026-04-03
> > >
> > > Thank you for the thoughtful follow-up and positive assessment of our work. We appreciate your engagement and are glad our rebuttal addressed your concerns.

---

### Official Review · Reviewer_gKoT · 2026-03-13

**Soundness:** 2
**Presentation:** 3
**Significance:** 3
**Originality:** 2
**Overall Recommendation:** 4
**Confidence:** 2

**Summary:**

This paper investigates how to extend the supervision mechanism for code repair by utilizing unit tests as the sole validator. The authors propose a generator-fixer self-game model where a single LLM alternately performs the following tasks: (i) generating defective programs through unconstrained text editing; and (ii) fixing these programs. The model is trained using GRPO with pass/fail as a reward and further tweaked based on the “difficulty” of the defects. They discover a key failure mode—distribution drift—where the self-game model learns to generate increasingly artificial but test-unsuccessful defects, thus improving at fixing self-generated defects but regressing at fixing real/human-generated defects. To mitigate this problem, they introduce ASP, which anchors training to a small set of reference defects by: (a) mixing reference defects into the fixer training; and (b) using embedding-based k-nearest neighbor similarity to guide the generator to generate edits similar to the reference defects through similarity-guided rewards. They also introduced BugSourceBench, a controlled benchmark used to evaluate fixes across multiple defect sources, varying the distribution of defect sources while keeping the task constant.

**Compliance With Llm Reviewing Policy:**

Affirmed.

**Final Justification:**

The rebuttal solved my problem, but the overall quality of the manuscript still makes me maintain a weak acceptance.

**Key Questions For Authors:**

1. How does performance vary with the size and composition of the reference set? Is there a minimum reference set size beyond which performance gains tend to saturate?
2. The generator reward uses K repair attempts to estimate the difficulty. Please report the total computational cost and comparisons of computational cost matching; otherwise, some performance gains may be attributable to more evaluation attempts per training step.
3. Even at a smaller scale, do you observe the same drift and ASP gain in external benchmarks or different task families?

**Limitations:**

Partly so. This paper discusses the limitations of distribution drift and reference anchoring and a brief section on "Wideer Limitations" is recommended.

**Strengths And Weaknesses:**

Strengths:
1. This paper clearly reveals an important and plausible failure mode in the open generator-fixer self-game: test-based reward mechanisms fail to enforce authenticity, leading to distribution drift.
2. The proposed mitigation measures (anchoring) are technically sound and consistent with the diagnosed problem: reference mixing reduces overfitting to the evolving generator distribution, while similarity shaping directly biases the generator toward the target distribution.
3. The evaluation design is an effective methodological choice that avoids confounding task difficulty and makes the explanation of distribution drift more convincing.

Weaknesses:
1. Similarity rewards rely on a proprietary embedding model and a specific representation. It remains unclear how robust the results are to factors such as embedding selection, k-nearest neighbor pooling, and whether the reward mechanism encourages superficial similarity rather than semantic authenticity. Ablation using other embeddings, or at least increasing the sensitivity to k/embedding models, can enhance model reliability.
2. The generator's reward is based on a "difficulty range" estimated using K repair attempts. This introduces additional randomness and computational cost; the paper should more explicitly analyze whether the improvement stems from better error realism, simply better difficulty calibration, or increased computational cost in evaluating each error.
3. Training uses a single model that shares weights between the generator and repairer. It is not entirely clear whether the observed drift and anchoring effects persist if the generator/repairer is decoupled, or if the generator is significantly stronger/weaker than the repairer—this is crucial for the model's generality.

---

> ### Author Rebuttal · Authors · 2026-03-31
>
> Thank you for your review and constructive feedback! We address your concerns below, and have revised our paper in response to your comments.
>
> > (W1) Embedding selection, kNN pooling, and superficial vs. semantic similarity
>
> We ran ablations over both the embedding model and k. We find that ASP is not tied to one embedding model and is stable across moderate k:
>
> |Embedding|Avg Fix Rate (%)|
> |:--|--:|
> |voyage-code-3 (default)|36.1|
> |**CodeBERT**|**36.4**|
>
>
> |k (voyage-code-3)|Avg Fix Rate (%)|
> |:--|--:|
> |1|34.7|
> |5 (default)|36.1|
> |10|35.8|
> |20|35.0|
>
> ASP’s gains hold across two very different embedding models, voyage-code-3 and CodeBERT, suggesting that the reward captures meaningful distributional structure rather than superficial similarity.
>
> Moreover, Figure 7a of our paper shows that different bug sources have distinct semantic profiles (e.g., human edits skew toward logic errors, Qwen-7B toward API misuse). Figure 7b shows that the embedding space recovers these source-level clusters. This suggests that nearby embeddings reflect semantically similar bug patterns.
>
> > (W2/Q2) Computational cost; realism vs. difficulty calibration vs. compute.
>
> ASP and vanilla self-play are exactly compute matched: both use the same rollout protocol (G=4 candidate bugs and K=4 repair attempts, yielding 16 fixer rollouts per task), the same training horizon (120 steps), and the same hardware (8 H100s).  Fixer-only is likewise matched in repair budget, using n=16 repair attempts per task.
>
> Both ASP and vanilla self-play share the same difficulty-band reward, so the only difference is anchoring. Table 1a confirms both ASP components independently improve fix rate. Figure 9b shows ASP outperforms self-play even within matched similarity bins, suggesting gains stem from better error realism.
>
> > (W3) Does drift and ASP gains persist with decoupled generator/fixer or asymmetric strength between 2 roles?
>
> We ran experiments where we decoupled weights between the 2 roles.
>
> |Method|Weights|Avg|Human|Human-Ed.|Qwen-7B|gpt-oss-20b|
> |---|---|---:|---:|---:|---:|---:|
> |Self-Play|shared|20.9|31.7|24.4|12.6|15.0|
> |Self-Play|decoupled|19.6|28.5|22.5|12.0|15.5|
> |ASP (Ours)|shared|35.5|64.2|37.0|18.9|22.0|
> |ASP (Ours)|decoupled|34.8|61.5|36.0|18.0|23.5|
>
> Drift is *worse* with decoupled weights (human: 28.5 vs 31.7), while ASP's gains are robust to decoupling (34.8 vs 35.5). This confirms drift is a property of the reward signal, not of weight sharing.
>
> For asymmetric strength: fixer-only (frozen base model as generator) already tests this setting and shows limited gains on human bugs (Figure 3), motivating ASP's co-evolving but anchored generator.
>
> > (Q1) How does performance vary with reference set size and composition?
>
> These results are in the paper: Figure 5 shows ASP improves with as few as 50 reference examples, and continues improving up to 900 with no clear saturation in the tested range. Table 1b shows that composition matters: human-only references perform best on human bugs, LM-only references shift gains toward LM-originated bugs, and the mixed pool used by ASP gives the best overall average.
>
> > (Q3) Drift and ASP gains on external benchmarks
>
> We ran ASP and Self-Play on DebugBench [(Tian et al., ACL 2024)](https://arxiv.org/abs/2401.04621), a benchmark of bugs in LeetCode code snippets, which represents a different task family and distribution. We present fix rate on DebugBench bugs for a held-out set of problems.
>
> |Model|Fix Rate|
> |---|---|
> |Qwen-7B|58.0%|
> |Self-Play|58.7%|
> |ASP (Ours)|66.3%|
>
> Self-play exhibits drift on DebugBench held-out problems while ASP stably improves, confirming drift is a fundamental property of unit-test-only self-play, not specific to BigCodeBench.
>
> In transfer (BigCodeBench-trained models evaluated on DebugBench without retraining), ASP also outperforms self-play, suggesting better generalizability to OOD bug distributions.
>
> |Model|Fix Rate|
> |---|---|
> |Qwen-7B|55.0%|
> |Fixer-Only|54.6%|
> |Self-Play|58.7%|
> |ASP (Ours)|60.3%|
>
> > (L) Wider limitations section
>
> Thank you for the suggestion. We’ve added a broader limitations section and mention limitations regarding programming languages and repo-level settings.

---

> > ### Author Rebuttal · Reviewer_gKoT · 2026-04-03
> >
> > Thank you for your reply.

---

> > > ### Author Response · Authors · 2026-04-03
> > >
> > > Thank you for taking the time to review our rebuttal. We're glad our responses addressed your concerns, and we sincerely appreciate your constructive feedback and engagement with our work.
> > >
> > > We also wanted to follow up on one point: since you selected option (a), indicating that your concerns were fully resolved, we wondered whether you had an opportunity to reconsider your score. We understand if you prefer to keep your current evaluation, but we wanted to make sure there are no remaining concerns.

---

### Official Review · Reviewer_GjTf · 2026-04-06

**Soundness:** 3
**Presentation:** 3
**Significance:** 3
**Originality:** 3
**Overall Recommendation:** 4
**Confidence:** 3

**Summary:**

This paper studies whether unit tests alone can provide a sufficiently strong signal to support self-play for code repair. The proposed framework uses a generator–fixer setup and introduces **ASP**, which adds a similarity-based reward to guide bug generation toward more realistic edits and mixes a small number of reference bugs into fixer training to reduce distribution drift. The paper also introduces **BUGSOURCEBENCH**, a benchmark with human-written, human-edited LM, and LM-generated bugs. Overall, the paper presents a clear and well-motivated attempt to improve the realism and transferability of self-play-based code repair.

**Compliance With Llm Reviewing Policy:**

Affirmed.

**Final Justification:**

Overall, the paper presents a clear problem formulation, a well-motivated method, and an experimental setup that is largely sufficient to support its main claims. While the degree of novelty and completeness of validation could be further strengthened, the current limitations do not outweigh the practical and empirical value of the contribution. The paper offers a useful step forward on self-play for code repair and distributional robustness across bug sources.

**Key Questions For Authors:**

1. Could you provide a small amount of **stability evidence**, such as multiple seeds for the main result or a short note on training variance?
2. Can you further justify the use of **embedding similarity** as a realism signal, either conceptually or with a lightweight validation experiment?
3. Can you clarify more explicitly how ASP differs from prior synthetic-bug learning approaches, especially in terms of problem setting and intended contribution?
4. How well do you expect the method to perform when the available reference bugs are less aligned with the test distribution?

**Limitations:**

yes

**Strengths And Weaknesses:**

### Strengths

* **Technical novelty and innovation:**
  The paper identifies an important issue in test-only self-play, namely that bugs can be test-valid without being realistic. This is a meaningful observation, and the proposed anchoring strategy is simple, intuitive, and practically useful.

* **Experimental rigor and validation:**
  The evaluation is thoughtfully structured. In particular, varying the **source of bugs** while keeping the repair setting fixed is a strong design choice. The paper also includes useful ablations and additional analyses that help clarify the role of each component.

* **Clarity of presentation:**
  The method is generally well explained. The roles of the generator and fixer are clearly separated, and the paper communicates the motivation, setup, and findings in a direct and understandable way.

* **Significance of contributions:**
  The paper addresses a relevant problem for code intelligence and self-improving training pipelines. The benchmark contribution is also valuable, especially because it encourages evaluation across different bug sources rather than only on one synthetic distribution.

### Weaknesses

* **Limited validation of the realism signal:**
  The central idea of ASP relies on embedding-based similarity as a proxy for realism. This is reasonable, but the paper would be stronger with one additional validation step, such as human judgment, an alternative similarity metric, or a short robustness analysis showing that the gains do not depend too heavily on one particular embedding choice.

* **Evaluation breadth could be slightly stronger:**
  The experimental results are promising, but the empirical case would be more complete with either multi-seed reporting or one additional training backbone. This is not a fatal issue, but some evidence of stability would make the conclusions more convincing.

* **Positioning against prior synthetic-bug frameworks could be clearer:**
  The paper is well motivated, but its relation to prior work such as BIFI, BUGLAB, or similar synthetic-bug learning paradigms could be stated more explicitly. Even a clearer discussion of what is inherited versus what is new would help sharpen the paper’s contribution.

---

### Decision · Program_Chairs · 2026-04-30

**Decision:**

Accept (regular)

**Comment:**

This paper studies how to scale supervision for code repair via self-play. The authors identify a key failure mode: vanilla self-play suffers from distribution drift, where it learns to produce increasingly unnatural and unrealistic bugs that simply break the tests, causing the fixer to overfit to synthetic patterns and degrade on real-world bugs. To mitigate this, the authors propose Anchored Self-Play (ASP), a method that uses a small set of real-world reference bugs to anchor the self-play process via two mechanisms: (1) a code-embedding similarity reward for the generator, and (2) reference mixing during the fixer's training. Additionally, the authors introduce BUGSOURCEBENCH, a rigorously designed benchmark that controls for task difficulty by using identical prompts and tests, but varies the source of the bugs (human-written, human-edited LM, and LM-generated). Extensive experiments demonstrate that ASP effectively prevents distribution drift and improves the repair rate across heterogeneous bug sources.

The overall presentation and evaluation of the method is good. Several concerns were raised by the reviewers, regrading robustness and generalizability of the method, limitation to function-level code repair and basline comparisons. However, most of them have been addressed in the rebuttal.

Authors, please update the paper according to the discussion with the reviewers in the final version.